# Attention Sink Forges Native MoE in Attention Layers: Sink-Aware Training to Address Head Collapse

**Zizhuo Fu** [1 2]   **Wenxuan Zeng** [1]   **Runsheng Wang** [2]   **Meng Li** [1 2]

## Abstract

Large Language Models (LLMs) often assign disproportionate attention to the first token, a phenomenon known as the *attention sink*. Several recent approaches aim to address this issue, including *Sink Attention* in GPT-OSS and *Gated Attention* in Qwen3-Next. However, a comprehensive analysis of the relationship among these attention mechanisms is lacking. In this work, we provide both theoretical and empirical evidence demonstrating that the *sink* in *Vanilla Attention* and *Sink Attention* naturally construct a Mixture-of-Experts (MoE) mechanism within attention layers. This insight explains the head collapse phenomenon observed in prior work, where only a fixed subset of attention heads contributes to generation. To mitigate head collapse, we propose a sink-aware training algorithm with an auxiliary load balancing loss designed for attention layers. Extensive experiments show that our method achieves effective head load balancing and improves model performance across *Vanilla Attention*, *Sink Attention*, and *Gated Attention*. We hope this study offers a new perspective on attention mechanisms and encourages further exploration of the inherent MoE structure within attention layers.

## 1. Introduction

StreamingLLM (Xiao et al., 2023) revealed that large language models (LLMs) often assign high attention weight to the first token, regardless of its semantic relevance. This phenomenon, termed *attention sink*, has been widely applied in various domains, including KV cache optimization (Ge et al., 2023; Wu & Tu, 2024; Su & Yuan, 2025), long-context generation (Xiao et al., 2024; Fu et al., 2025), attention head

pruning (Sandoval-Segura et al., 2025; Shin et al., 2025), and quantization (Liu et al., 2024; Huang et al., 2024).

Recent works have studied attention sink and explained its underlying causes (Guo et al., 2024; Gu et al., 2024; Barbero et al., 2025; de Llano et al., 2025). To mitigate this issue, researchers have proposed a range of methods that fall into two main categories, as illustrated in Figure 1 (b)(c). The first category modifies the softmax operation (Miller, 2023; Gu et al., 2024; Zuhri et al., 2025), represented by *Sink Attention* in GPT-OSS (Agarwal et al., 2025). The second category introduces a gating mechanism (Bondarenko et al., 2023), represented by *Gated Attention* in Qwen3-Next (Qiu et al., 2025b). Despite their promising empirical results, the mechanism behind the gains from eliminating attention sink is still unclear. Furthermore, a unified perspective connecting these different attention mechanisms and their relationship to attention sink remains underexplored.

In this paper, we present a comprehensive analysis of *Vanilla Attention*, *Sink Attention*, and *Gated Attention*. Our analysis reveals a surprising finding: the attention weight on the *sink* in *Vanilla Attention* and *Sink Attention* functions as an implicit gating factor, which corresponds directly to the explicit gating factor in *Gated Attention*, effectively routing each head like an expert in a Mixture-of-Experts (MoE) layer, as illustrated in Figure 1. We further demonstrate the advantages of *Sink Attention* and *Gated Attention* over *Vanilla Attention* after eliminating attention sinks. Specifically, these mechanisms enable more flexible and precise retrieval of relevant tokens via the softmax operation, which is particularly crucial in long-context scenarios.

Furthermore, we show that in existing LLMs, only a fixed subset of heads consistently contributes to the generation, while the others remain inactive. Similar observations have been reported in previous studies (Xiao et al., 2024; Sandoval-Segura et al., 2025; Sok et al., 2026). Building on our analysis that attention layers exhibit a native MoE structure, this phenomenon aligns with the well-known issue of expert collapse in MoE models (Shazeer et al., 2017; Fedus et al., 2021), where the model repeatedly relies on a small number of dominant experts. We term this phenomenon *head collapse*, where only a small subset of heads remain highly activated while others exhibit low activation.

---

[1]Institute for Artificial Intelligence, Peking University, Beijing [2]School of Integrated Circuits, Peking University, Beijing. Correspondence to: Meng Li <meng.li@pku.edu.cn>.

*Proceedings of the 43$^{rd}$ International Conference on Machine Learning*, Seoul, South Korea. PMLR 306, 2026. Copyright 2026 by the author(s).

Figure 1. **Vanilla Attention** used in most open-source models. **Sink Attention** used in GPT-OSS with a learnable bias *sink* added to the softmax denominator. **Gated Attention** used in Qwen3-Next with a head-wise gating factor computed via sigmoid activation.

To address the collapse issue and improve head load balancing, we propose a sink-aware auxiliary load balancing loss for all three attention mechanisms. It leverages the implicit gating factors in *Vanilla Attention* and *Sink Attention*, as well as the explicit gating factors in *Gated Attention*. For models trained from scratch, the loss encourages load balancing across all heads in attention layers. For fine-tuning existing pretrained LLMs where collapse has already emerged, we adopt a tailored approach that keeps the top activated heads as shared heads while applying load balancing only to the remaining routed heads, preventing disruption to already learned behaviors. Our implementation integrates seamlessly with Flash Attention (Dao, 2023) by leveraging log-sum-exp (LSE) values, enabling efficient training without explicitly extracting attention weights.

We conduct extensive experiments to validate our analysis and approach. Specifically, we train models of 0.6B, 1B, and 2B parameters from scratch under six configurations: *Vanilla Attention*, *Sink Attention*, and *Gated Attention*, each with and without the auxiliary load balancing loss. Additionally, we apply our method to fine-tune existing LLMs including LLaMA3 and Qwen3. The experimental results demonstrate that our approach effectively encourages balanced utilization across attention heads and leads to consistent improvements in model performance.

## 2. Preliminary

### 2.1. Vanilla Attention and Attention Sink

In Transformer attention layers (Vaswani et al., 2017), for the $h$-th head in the $l$-th layer, the attention weight and output are computed as:

$$A_{t,j}^{l,h} = \text{Softmax}\left(\frac{\mathbf{q}_t^{l,h}\mathbf{k}_j^{l,h\top}}{\sqrt{d_h}}\right), \quad O_t^{l,h} = \sum_{j=0}^{t} A_{t,j}^{l,h} \cdot \mathbf{v}_j^{l,h}$$

where $\mathbf{q}_t^{l,h}$, $\mathbf{k}_t^{l,h}$, and $\mathbf{v}_t^{l,h}$ denote the query, key, and value vectors for the $t$-th token, $A_{t,j}^{l,h}$ represents the attention weight from token $t$ to token $j$, $d_h$ is the head dimension, and $O_t^{l,h}$ is the output.

The attention sink arises because softmax normalization forces attention weights to sum to one (Xiao et al., 2023; Miller, 2023). When a head requires only partial or no attention allocation, redundant attention weight emerges. The model absorbs this redundancy by directing it to the first token, whose value vector is near zero due to minimal semantic content (Gu et al., 2024; Guo et al., 2024). This redundant attention, while satisfying normalization constraints, does not contribute meaningfully to the output.

### 2.2. Methods for Eliminating Attention Sink

**Sink Attention.** GPT-OSS (Agarwal et al., 2025) modifies the softmax denominator with a learnable parameter *sink*:

$$\text{Softmax}_{\text{sink}}(x)_i = \frac{\exp(x_i)}{\sum_j \exp(x_j) + \exp(\mathbf{sink})} \quad (1)$$

which relaxes the normalization constraint and eliminates attention sink by introducing a synthetic *sink* with zero value, replacing the first token in *vanilla attention*.

**Gated Attention.** *Gated Attention* (Qiu et al., 2025b) introduces two key designs: **a head-specific gating factor** and **sigmoid activation**. The gating factor for each head is computed as:

$$G_t^{l,h} = \sigma\left(\mathbf{x}_t^l W_\theta^{l,h}\right) \quad (2)$$

where $\sigma(\cdot)$ is the sigmoid function and $W_\theta^{l,h} \in \mathbb{R}^{d_{\text{model}} \times 1}$ is a learnable projection. The final output is computed as:

$$O_t^l = \sum_{h=1}^{H} \left(G_t^{l,h} \cdot O_t^{l,h}\right) W_O^{l,h}$$

where $W_O^{l,h} \in \mathbb{R}^{d_{\text{head}} \times d_{\text{model}}}$ is the output projection matrix for head $h$, and $O^l$ is the output of attention layer $l$. This head-specific gating mechanism effectively introduces an MoE structure within attention layers, enabling sparse head utilization and eliminating the sink phenomenon.

# 3. Attention Sink Forges Native MoE in Attention Layers

## 3.1. Implicit Gating Mechanism in Attention Variants

In this section, we reveal a unified perspective on *Vanilla Attention*, *Sink Attention*, and *Gated Attention*. Our analysis shows that the attention weight allocated to the sink (first token in *Vanilla Attention* and *sink* parameter in *Sink Attention*) functions as an implicit gating factor, corresponding to the explicit gating factor in *Gated Attention*. This insight establishes that attention sink naturally forges a native MoE structure within attention layers.

**Consistency between Vanilla Attention and Sink Attention.** We first examine how *Vanilla Attention* and *Sink Attention* handle redundant attention weight in a consistent manner. As discussed in Section 2, the softmax normalization forces attention weights to sum to one, leading to redundant attention weight when full allocation is unnecessary. Both mechanisms absorb this redundancy through a sink component whose value vector contributes negligibly to the output.

In *Vanilla Attention*, this is achieved through the *value drain* phenomenon, where the value vector of the first token approaches zero (Guo et al., 2024; Gu et al., 2024). Figure 2 illustrates this phenomenon across multiple models. The $\ell_2$-norm of the first token's value vector is consistently near zero compared to subsequent tokens, indicating that the model learns to nullify this value during training. This behavior arises because the first token receives substantial attention weight to absorb redundancy, and setting its value close to zero ensures that this attention does not affect the output. Additional experiments in Appendix A further validate this observation by showing that explicitly zeroing the first token's value yields negligible performance loss.

In *Sink Attention*, the learnable parameter *sink* replaces the first token as the absorption target for redundant attention. Since the *sink* parameter has no associated value vector, attention allocated to it contributes exactly zero to the output.

Therefore, both *Vanilla Attention* and *Sink Attention* employ the same underlying principle: directing redundant attention to a component with zero or near-zero value contribution.

**Implicit Gating in Vanilla and Sink Attention.** We now demonstrate that *Vanilla Attention* and *Sink Attention* possess an implicit gating mechanism that corresponds to the explicit gating in *Gated Attention*, as illustrated in Figure 1.

For notational clarity, we use $A_{t,\text{sink}}^{l,h}$ to denote the attention weight on the sink component, which equals $A_{t,0}^{l,h}$ in *Vanilla Attention* or $\frac{\exp(\mathbf{sink})}{\sum_j \exp(x_j) + \exp(\mathbf{sink})}$ in *Sink Attention*. Since the sink component has zero value contribution, it does not affect the output, i.e., $A_{t,sink}^{l,h} \cdot \mathbf{v}_{sink}^{l,h} \approx \mathbf{0}$. Thus, the

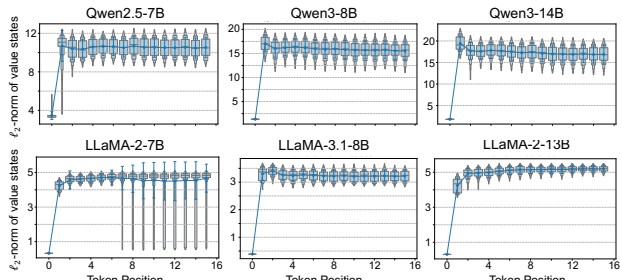

*Figure 2.* $\ell_2$-norm of token values across models. The first token's value vector approaches zero in models with *Vanilla Attention*.

effective output becomes:

$$O_t^{l,h} = \sum_{j \neq \text{sink}} A_{t,j}^{l,h} \cdot \mathbf{v}_j^{l,h} = (1 - A_{t,\text{sink}}^{l,h}) \cdot \sum_{j \neq \text{sink}} \tilde{A}_{t,j}^{l,h} \cdot \mathbf{v}_j^{l,h}$$

(3)

where $\tilde{A}_{t,j}^{l,h}$ denotes the re-normalized softmax attention weights over non-sink tokens, defined as $\tilde{A}_{t,j}^{l,h} = A_{t,j}^{l,h}/(1 - A_{t,\text{sink}}^{l,h})$ for $j \neq$ sink. The term $(1 - A_{t,\text{sink}}^{l,h})$ naturally emerges as a scaling factor on the weighted sum of values. Detailed derivations are provided in Appendix B.

In *Gated Attention*, the output is computed as $O_t^{l,h} = G_t^{l,h} \cdot \sum_{j=0}^{t} A_{t,j}^{l,h} \cdot \mathbf{v}_j^{l,h}$, where $G_t^{l,h}$ is the explicit gating factor from Equation (2). Comparing this with Equation (3), we identify that the implicit gating factor in *Vanilla Attention* and *Sink Attention* takes the form:

$$G_t^{l,h} = 1 - A_{t,\text{sink}}^{l,h}$$

(4)

This implicit gating factor aligns closely with the two central design choices of *Gated Attention* (Qiu et al., 2025b). First, it is head-specific: each attention head computes its own sink attention weight, yielding independent gating across heads. Second, it has an equivalent sigmoid form, aligning with Qiu et al. (2025b), where sigmoid is empirically found to work best for gating. For *Sink Attention*, we can rewrite it as $G_t^{l,h} = \sigma(\text{LSE} - \mathbf{sink})$, where LSE is the log-sum-exp of the pre-softmax logits. For *Vanilla Attention*, this becomes $G_t^{l,h} = \sigma\left(\text{LSE}_{\neg 0} - \frac{\mathbf{q}_t^{l,h}\mathbf{k}_0^{l,h\top}}{\sqrt{d_h}}\right)$, where $\text{LSE}_{\neg 0}$ excludes the first token. Detailed proofs are provided in Appendix B.

**Attention Sink Forges Native MoE.** The MoE framework comprises two essential components: independent expert modules and a routing mechanism that governs their contributions (Shazeer et al., 2017; Fedus et al., 2021). Attention layers naturally satisfy these requirements: Multiple attention heads serve as independent experts, with the implicit gating factor derived from attention sink providing the routing mechanism. Therefore, attention sink forges a native MoE structure within attention layers.

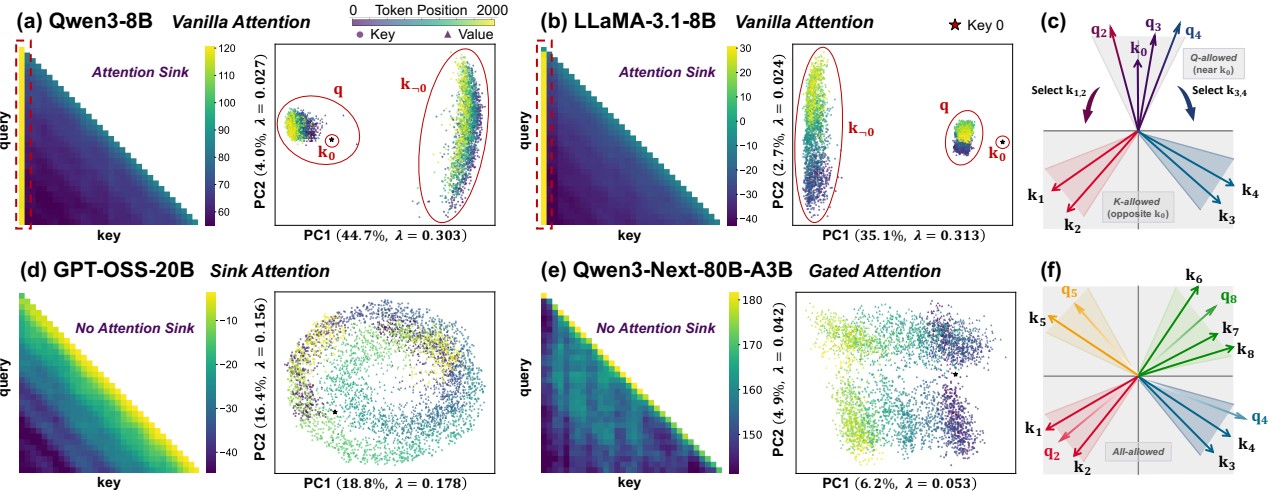

*Figure 3.* Visualization of attention patterns and query-key geometry. Each panel shows the attention map (left) and PCA projection of query and key vectors (right), where red stars indicate $k_0$. (a)(b)(c) *Vanilla Attention* models exhibit attention sink and constrained query-key geometry. (d)(e)(f) *Sink Attention* and *Gated Attention* models show no sink phenomenon and more flexible vector distributions.

### 3.2. Advantages of Eliminating Attention Sink

Having established the consistency among attention mechanisms, we now analyze the advantages of *Sink Attention* and *Gated Attention* over *Vanilla Attention*. By decoupling the sink from semantic tokens, these mechanisms enhance the expressiveness of query-key matching.

**Limitations of Using the First Token as Sink.** Prior work has identified several negative effects of attention sink. First, attention sink induces extreme outliers in activation values (Xiao et al., 2022; Bondarenko et al., 2023; Zhao et al., 2023), which complicates quantization. Second, attention sink degrades model expressiveness (Gu et al., 2024; Zuhri et al., 2025). Eliminating attention sink consistently improves performance on long-context benchmarks (Ramapuram et al., 2024; Qiu et al., 2025b). However, these studies do not explain why removing attention sink enhances expressiveness, especially in long-context scenarios.

Our analysis reveals the underlying cause. Figure 3 visualizes the attention patterns and query-key geometry across different attention mechanisms. We employ PCA-based 2D scatter plots to illustrate the geometric distribution and relative distances of query and key vectors, using samples of 2k tokens from LongBench (Bai et al., 2023).

In *Vanilla Attention* models (Figure 3(a)(b)(c)), using the first token as the attention sink imposes geometric constraints that produce a clear bipolar structure. To maintain high attention weight on the first token, query vectors of subsequent tokens must cluster near $k_0$, while key vectors of these tokens are pushed away from $k_0$ to prevent interference with the sink mechanism. This polarization severely limits the representational capacity of the attention mechanism. Query vectors can only vary within a narrow angular

range, and the effective dimensionality of key vectors is similarly constrained. Consequently, the softmax operation loses precision in distinguishing among candidate tokens, as the relative differences in query-key dot products become compressed. In long-context scenarios, this imprecision introduces substantial attention noise (Liu et al., 2023), degrading the model's ability to retrieve relevant information.

**Enhanced Flexibility in Sink and Gated Attention.** In contrast, *Sink Attention* and *Gated Attention* models (Figure 3(d)(e)(f)) exhibit different behavior. The first token's key vector $k_0$ no longer occupies a special position, and both query and key vectors display diverse, distributed patterns without pronounced polarization. This flexibility enables more accurate query-key matching and finer-grained token selection. The softmax operation can leverage the full representational capacity to distinguish among tokens, which is particularly beneficial for long-context scenarios.

## 4. Sink-Aware Training to Address Head Collapse

### 4.1. Head Collapse in Attention Layers

Prior work reveals that only a fixed subset of attention heads actively contributes to generation. DuoAttention (Xiao et al., 2024) shows that important heads remain consistent across datasets, enabling efficient inference by applying full attention only to important heads. More strikingly, Sandoval-Segura et al. (2025) and Sok et al. (2026) demonstrate that zeroing out over 25% of heads preserves model accuracy, as these dormant heads contribute minimally to any task. This pattern resembles *expert collapse* phenomenon in MoE models (Shazeer et al., 2017; Fedus et al., 2021), where the model relies disproportionately on a fixed subset of experts.

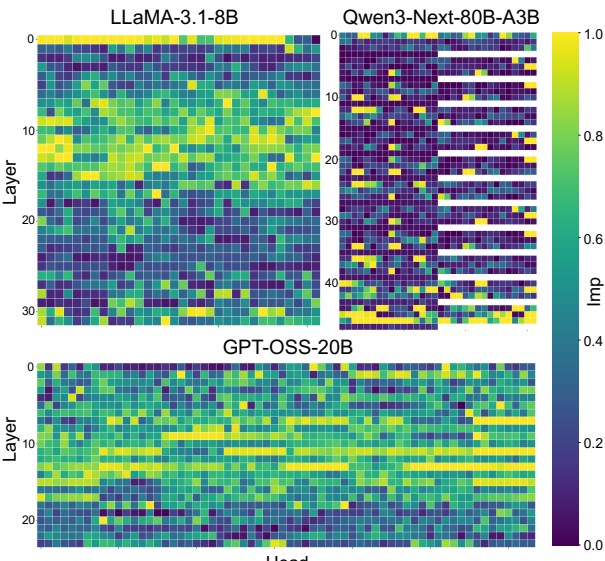

*Figure 4.* Head importance scores over all tokens and samples for LLaMA-3.1-8B (*Vanilla Attention*), GPT-OSS-20B (*Sink Attention*), and Qwen3-Next-80B-A3B (*Gated Attention*). Qwen3-Next-80B-A3B uses different numbers of heads in different layers.

**Quantifying Head Importance.** Building on our analysis in Section 3.1 that attention layers exhibit a native MoE structure, we can now examine head collapse through the lens of gating mechanisms. The implicit and explicit gating factors derived in the previous section provide a natural way to quantify each head's contribution to generation. We define the importance score of head $h$ in layer $l$ as the mean gating factor across all tokens in a batch:

$$\text{Imp}^{l,h} = \frac{1}{|\mathcal{T}|} \sum_{(i,t) \in \mathcal{T}} G_{i,t}^{l,h} \quad (5)$$

where $\mathcal{T}$ indexes all tokens in the batch, where $i$ denotes the sample index and $t$ denotes the token position. $|\mathcal{T}|$ is the total number of batched tokens. We use a unified definition of $G_{i,t}^{l,h}$ across attention variants. For Gated Attention, $G_{i,t}^{l,h}$ is the sigmoid gate in Equation (2). For Vanilla Attention and Sink Attention, this corresponds to $1 - A_{sink}^{l,h}$ in Equation (4). Higher $\text{Imp}^{l,h}$ indicates that head $h$ is more strongly activated and contributes more to the layer output.

Figure 4 visualizes the importance scores across heads for three representative LLMs. The heatmaps reveal a stark pattern of head collapse across all three attention mechanisms. In each model, certain heads consistently exhibit high importance scores, while many others show very low activation levels. This pronounced disparity demonstrates that head collapse is a universal phenomenon affecting all three attention variants. See Appendix C for more results.

**Measuring Head Imbalance.** To quantify the severity of head collapse across different models and training stages, we define a metric that measures head load imbalance within each layer. Specifically, we compute the coefficient of variation (CV) of importance scores across heads in each layer, then average these values over all layers:

$$\text{Head Imbalance} = \frac{1}{N_L} \sum_{l=1}^{N_L} \text{CV}\left(\{\text{Imp}^{l,h}\}_{h=1}^{N_h}\right) \quad (6)$$

where $N_L$ is the number of layers, $N_h$ is the number of heads per layer, and $\text{CV}(\cdot) = \frac{\text{std}(\cdot)}{\text{mean}(\cdot)}$ denotes the coefficient of variation. Higher values indicate greater imbalance in head utilization.

Figure 5 tracks the head load imbalance metric throughout training for models of varying sizes using three different attention mechanisms. The plots reveal a consistent trend: head collapse intensifies as training progresses. Across all model sizes and attention variants, the imbalance metric increases during early training stages and eventually plateaus at elevated levels. This suggests that head collapse emerges naturally during optimization, as certain heads become dominant early in training and subsequently attract disproportionate updates, while other heads remain underutilized.

**Impact of Head Collapse.** The consequences of head collapse are multifaceted and detrimental. First, it leads to inefficient utilization of model capacity. When a substantial fraction of attention heads contributes minimally to generation, the effective parameter count is far below the nominal model size. Second, as demonstrated in prior work on expert collapse in MoE models (Shazeer et al., 2017; Fedus et al., 2021), imbalanced utilization can degrade model performance by preventing the model from learning diverse and specialized representations. Third, collapsed heads represent a significant waste of computational resources. Both training and inference incur costs for computing attention outputs from all heads, yet many of these computations produce negligible contributions to the final result.

### 4.2. Sink-Aware Auxiliary Load Balancing Loss

To mitigate head collapse and improve head utilization, we introduce a sink-aware auxiliary load balancing loss specifically designed for attention layers. Our approach leverages the implicit gating factors in *Vanilla Attention* and *Sink Attention*, as well as the explicit gating factor in *Gated Attention*, drawing inspiration from load balancing techniques in MoE models (Shazeer et al., 2017; Fedus et al., 2021).

**Auxiliary Loss for Training from Scratch.** For models trained from scratch, we encourage balanced head utilization by minimizing the coefficient of variation of importance scores within each layer. The auxiliary load balancing loss is defined as:

$$\mathcal{L}_{\text{aux}} = \lambda \sum_{l=1}^{N_L} N_h \left[ \text{CV}\left(\{\text{Imp}^{l,h}\}_{h=1}^{N_h}\right) \right]^2 \quad (7)$$

where $N_L$ is the number of layers, $N_h$ is the number of heads per layer, and $\lambda$ is a hyperparameter controlling the strength of the auxiliary loss. This loss encourages uniform importance scores across all heads within each layer, promoting balanced head utilization throughout training. The total training objective becomes $\mathcal{L} = \mathcal{L}_{\text{base}} + \mathcal{L}_{\text{aux}}$, where $\mathcal{L}_{\text{base}}$ is the standard language modeling loss.

**Adaptation to Fine-Tuning Existing Models.** While the auxiliary loss in Equation (7) is effective for training from scratch, directly applying it to fine-tune existing LLMs leads to a phenomenon we term the head pinning effect. During pretraining, collapsed heads become dominant and essential for generation, causing their gating factors to resist change. Consequently, other heads adapt by increasing their own gating factors to match the dominant ones, failing to achieve balanced utilization. See Appendix D for detailed analysis.

To address this issue, we draw inspiration from DeepSeek-MoE (Dai et al., 2024), which employs both shared experts and routed experts in each MoE layer. We adopt an analogous strategy for attention heads: we designate the top-$m$ most important heads in each layer as shared heads, which preserve their learned behaviors, while treating the remaining heads as routed heads subject to load balancing. Let $\mathcal{T}_m^l = \arg\text{top}_m\{\text{Imp}^{l,h}\}_{h=1}^{N_h}$ denote the set of top-$m$ heads in layer $l$. The refined auxiliary loss for fine-tuning is:

$$\mathcal{L}_{\text{aux}} = \lambda \sum_{l=1}^{N_L} (N_h - m) \left[ \text{CV}\Big(\{\text{Imp}^{l,h} : h \notin \mathcal{T}_m^l\}\Big) \right]^2 \quad (8)$$

**Efficient Implementation with Flash Attention.** A practical challenge arises when implementing sink-aware training with modern efficient attention kernels. The implicit gating factor $G_t^{l,h} = 1 - A_{t,\text{sink}}^{l,h}$ from Equation (4) requires access to attention weights, which are not exposed in fused kernels such as Flash Attention (Dao, 2023). Extracting these weights would necessitate switching to *eager* mode, sacrificing computational efficiency.

To maintain compatibility with Flash Attention, we leverage the log-sum-exp (LSE) values that are already computed during the forward pass. For *Vanilla Attention*, the implicit gating factor can be rewritten using the sigmoid formulation: $G_t^{l,h} = \sigma(\text{LSE}_{\neg 0} - \frac{\mathbf{q}_t^{l,h} \mathbf{k}_0^{l,h\top}}{\sqrt{d_h}})$, where $\text{LSE}_{\neg 0}$ denotes the log-sum-exp excluding the first token and $\sigma(\cdot)$ is the sigmoid function. By calling the Flash Attention kernel as *out, lse = torch.ops.aten._scaled_dot_product_flash_attention(q, k, v)*, we can compute $G_t^{l,h}$ from the returned LSE values and the query-key product for the first token, avoiding the need to materialize full attention weights.

For *Sink Attention*, the learnable parameter *sink* prevents direct use of standard Flash Attention implementations. Existing approaches (Agarwal et al., 2025) resort to un-

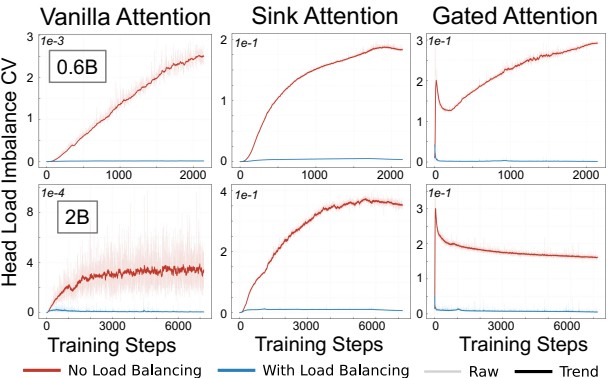

*Figure 5.* Head load imbalance during training across model scales and attention mechanisms. Each subplot shows the coefficient of variation (Equation (6)) over training steps for models trained with and without the auxiliary load balancing loss. Raw data and exponential moving average (EMA) smoothed trends are displayed.

fused operations, which significantly degrade training and inference efficiency. We propose an efficient alternative by leveraging the equivalence shown in Equation (4). Specifically, we compute attention outputs using standard Flash Attention over all tokens and then apply the gating factor $G_t^{l,h} = \sigma(\text{LSE} - \text{sink})$, where LSE is obtained from the Flash Attention kernel. Moreover, this approach directly provides the gating factors needed for the auxiliary load balancing loss, enabling seamless integration of our training method with modern optimization frameworks.

## 5. Experiments

### 5.1. Experimental Setups

**Base Model Training.** We train models from scratch with three attention mechanisms: *Vanilla Attention*, *Sink Attention*, and *Gated Attention*. We conduct experiments across three model scales: 0.6B, 1B, and 2B parameters. The 0.6B model consists of 20 layers with 10 attention heads per layer and a hidden dimension of 1280. The 1B model has 26 layers with 13 heads per layer and a hidden dimension of 1664. The 2B model comprises 32 layers with 16 heads per layer and a hidden dimension of 2048. We train all models on FineWeb-Edu (Penedo et al., 2024) and set the training token budget following the Chinchilla optimal scaling law (Hoffmann et al., 2022) to 20× the number of model parameters. We use a learning rate of 0.02 and linearly decay to 0 over the final 20% of training. We set $\lambda = 1 \times 10^{-4}$, and other hyperparameters follow the default values of the AdamW optimizer. As discussed in Section 4.2, our method integrates seamlessly with Flash Attention, resulting in less than 2% additional training latency.

We evaluate model performance on popular benchmarks, including MMLU (Hendrycks et al., 2020) for general knowledge, GSM8K (Cobbe et al., 2021) for mathemat-

*Table 1.* Evaluation results of 0.6B, 1B, and 2B models trained from scratch with different attention mechanisms and loss functions. *base* denotes training with the base loss only, while *base+aux* includes the auxiliary load balancing loss. ↑ indicates improved accuracy when adding the auxiliary load balancing loss.

| Attention Mechanism | Training Loss | Validation BPB (↓) | MMLU | GSM8K | HumanEval | ARC-E | ARC-C | HellaSwag | PIQA | OpenBookQA | BoolQ | Winogrande | Average |
|---|---|---|---|---|---|---|---|---|---|---|---|---|---|
| *Model Size = 0.6B* | | | | | | | | | | | | | |
| Vanilla | base | 0.8152 | 28.81 | 5.31 | 6.10 | 63.80 | 32.85 | 44.70 | 68.39 | 34.00 | 57.15 | 54.30 | 39.54 |
|  | base+aux | 0.8123 | 32.60 | 5.84 | 9.76 | 65.28 | 34.47 | 45.04 | 68.44 | 34.40 | 58.20 | 54.20 | 40.82 ↑0.28 |
| Sink | base | 0.8123 | 33.09 | 5.38 | 7.93 | 63.30 | 33.78 | 44.44 | 68.11 | 34.20 | 60.45 | 52.40 | 40.31 |
|  | base+aux | 0.8116 | 33.23 | 5.84 | 9.76 | 63.80 | 34.13 | 44.81 | 69.53 | 36.40 | 60.76 | 52.80 | 41.11 ↑0.80 |
| Gated | base | 0.8176 | 31.31 | 5.53 | 6.71 | 63.97 | 34.89 | 43.42 | 68.06 | 35.80 | 59.40 | 55.17 | 40.43 |
|  | base+aux | 0.8121 | 33.24 | 5.31 | 9.15 | 64.86 | 35.24 | 44.76 | 68.72 | 37.00 | 60.40 | 54.80 | 41.35 ↑0.92 |
| *Model Size = 1B* | | | | | | | | | | | | | |
| Vanilla | base | 0.7591 | 36.08 | 6.22 | 10.98 | 68.56 | 38.82 | 52.13 | 72.09 | 37.40 | 47.06 | 56.59 | 42.59 |
|  | base+aux | 0.7562 | 37.65 | 7.43 | 11.59 | 68.80 | 38.99 | 52.85 | 72.40 | 38.00 | 48.10 | 57.69 | 43.35 ↑0.76 |
| Sink | base | 0.7583 | 36.58 | 6.67 | 5.49 | 68.73 | 40.35 | 52.63 | 73.40 | 36.40 | 58.13 | 54.30 | 43.27 |
|  | base+aux | 0.7567 | 35.76 | 7.13 | 7.32 | 68.81 | 40.20 | 52.80 | 73.80 | 36.80 | 58.60 | 56.27 | 43.75 ↑0.48 |
| Gated | base | 0.7572 | 36.48 | 7.73 | 6.71 | 67.25 | 38.73 | 52.40 | 70.78 | 36.00 | 54.12 | 56.67 | 42.69 |
|  | base+aux | 0.7547 | 36.73 | 8.64 | 9.76 | 68.35 | 41.98 | 53.29 | 71.54 | 38.00 | 54.80 | 56.75 | 43.98 ↑1.29 |
| *Model Size = 2B* | | | | | | | | | | | | | |
| Vanilla | base | 0.7243 | 39.40 | 9.33 | 6.71 | 72.26 | 41.97 | 58.26 | 73.72 | 37.60 | 58.96 | 56.43 | 45.46 |
|  | base+aux | 0.7198 | 41.48 | 9.63 | 7.93 | 72.80 | 43.09 | 59.66 | 74.10 | 39.40 | 60.76 | 59.43 | 46.83 ↑1.37 |
| Sink | base | 0.7226 | 40.46 | 10.24 | 3.66 | 73.06 | 44.28 | 58.62 | 75.24 | 39.60 | 59.80 | 58.56 | 46.35 |
|  | base+aux | 0.7202 | 40.90 | 10.61 | 6.10 | 73.40 | 44.80 | 59.22 | 74.65 | 39.40 | 57.40 | 58.48 | 46.50 ↑0.15 |
| Gated | base | 0.7232 | 38.48 | 9.63 | 10.98 | 72.60 | 44.36 | 59.09 | 74.48 | 37.40 | 60.21 | 55.80 | 46.30 |
|  | base+aux | 0.7197 | 38.98 | 10.69 | 11.59 | 73.20 | 44.40 | 59.52 | 77.20 | 39.80 | 61.04 | 60.60 | 47.70 ↑1.40 |

ical reasoning, HumanEval (Chen et al., 2021) for coding, BoolQ (Clark et al., 2019) for reading comprehension, ARC-Easy, ARC-Challenge (Clark et al., 2018) and OpenBookQA (Mihaylov et al., 2018) for scientific reasoning, as well as HellaSwag (Zellers et al., 2019), PIQA (Bisk et al., 2019) and WinoGrande (Sakaguchi et al., 2019) for commonsense reasoning.

**Fine-Tuning on Existing LLMs.** We evaluate our method on three pretrained models: Qwen3-4B, Qwen3-8B (Yang et al., 2025), and LLaMA3.1-8B (Dubey et al., 2024). Fine-tuning is conducted on the AceReason-Nemotron dataset (Liu et al., 2025), a long-sequence reasoning dataset that allows us to assess the effectiveness of our auxiliary loss on both reasoning and long-context tasks. We adopt LoRA (Hu et al., 2021) for efficient fine-tuning with a rank of 16. During fine-tuning, we set $\lambda = 1 \times 10^{-2}$. All experiments are conducted on NVIDIA A100 GPUs.

We assess fine-tuned models on two sets of benchmarks. For reasoning ability (Qiu et al., 2025a), we use six datasets: ARC (Clark et al., 2018), MuSR (Sprague et al., 2023), GSM8K (Cobbe et al., 2021), MATH-500 (Hendrycks et al., 2021), Competition-Math (Hendrycks et al., 2021), and Process-Bench (Zheng et al., 2024). For long-context understanding, we use six tasks from LongBench (Bai et al., 2023): HotpotQA (Yang et al., 2018), Qasper (Dasigi et al., 2021), TriviaQA (Joshi et al., 2017), NarrativeQA (Kociský et al., 2017), 2WikiMultiHopQA (Ho et al., 2020), and MultiFieldQA (Bai et al., 2023).

### 5.2. Training from Scratch

Table 1 presents the evaluation results of models trained from scratch across different attention mechanisms and model scales. We observe several key findings. First,

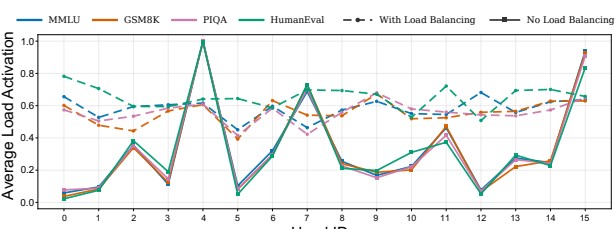

*Figure 6.* Head load activation across different datasets for 2B models trained with and without the auxiliary load balancing loss.

*Sink Attention* and *Gated Attention* consistently outperform *Vanilla Attention* across all model sizes. This improvement aligns with our analysis in Section 3.2, demonstrating that eliminating attention sink enhances model expressiveness not only on long-context tasks but also on general tasks. Among the three mechanisms, *Gated Attention* achieves the best performance across different model scales, validating the advantages of explicit gating mechanisms. Second, introducing the auxiliary load balancing loss yields consistent improvements across all attention mechanisms and model scales. The performance gains are evident in both downstream task accuracy and validation BPB. These results demonstrate that addressing head collapse through load balancing enhances model capacity.

Figure 5 illustrates the evolution of head load imbalance throughout training, measured by the coefficient of variation defined in Equation (6). The results reveal that head collapse emerges consistently across all attention mechanisms and model scales. Introducing the auxiliary load balancing loss effectively mitigates this phenomenon, maintaining more balanced head utilization throughout training.

Notably, different attention mechanisms exhibit distinct collapse patterns. *Gated Attention* shows a sharp increase in head load imbalance during early training, which then

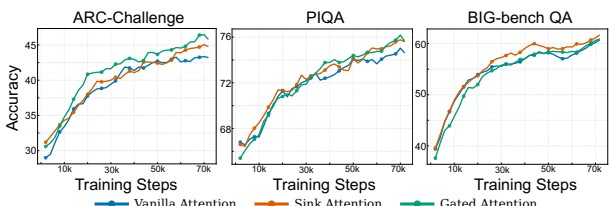

*Figure 7.* Model accuracy across datasets over training steps for different attention mechanisms.

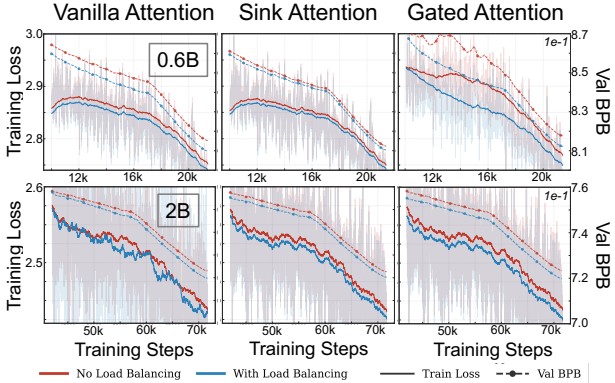

*Figure 8.* Training loss and validation BPB during training across model scales and attention mechanisms. Each subplot compares models trained with and without the auxiliary load balancing loss.

stabilizes at a plateau. In contrast, *Vanilla Attention* and *Sink Attention* display a gradual increase in imbalance over training steps. This difference arises from the nature of their gating mechanisms. In *Gated Attention*, the explicit gating factor enables immediate specialization of heads, leading to rapid collapse. For *Vanilla Attention* and *Sink Attention*, the implicit gating mechanism must first be established through attention sink allocation or the learnable *sink* parameter before head collapse can occur, resulting in a slower progression. Furthermore, *Vanilla Attention* exhibits pronounced oscillations compared to *Sink Attention*. This instability stems from the additional complexity of establishing value drain (Barbero et al., 2025; Gu et al., 2024), where the model must simultaneously learn to allocate attention to the first token and nullify its value contribution.

Figure 6 visualizes the average load activation $\mathrm{Imp}^{l,h}$ (Equation (5)) for heads within a layer across different datasets. Without load balancing, activation concentrates on a small subset of heads, and the pattern is highly consistent across datasets, indicating severe collapse. With load balancing, head activations become more even, and the patterns vary with the dataset, suggesting improved head utilization.

Figure 7 shows model accuracy as training progresses across datasets. Across training, *Sink Attention* and *Gated Attention* achieve higher accuracy than *Vanilla Attention*.

Figure 8 shows the training loss and validation BPB during training. In the early stage, adding the auxiliary load balancing loss has little effect. As training continues, models

*Table 2.* Evaluation results of models fine-tuned with different loss functions. The optimal results are in boldface.

| Dataset | Qwen3-4B | | Qwen3-8B | | LLaMA3.1-8B | |
|---|---|---|---|---|---|---|
| | base | base+aux | base | base+aux | base | base+aux |
| *Reasoning Task* | | | | | | |
| **ARC** | 89.98 | **90.84** | 91.98 | **92.34** | 80.41 | **80.60** |
| **MuSR** | 54.14 | **55.46** | 49.78 | **53.50** | 44.87 | **46.17** |
| **GSM8K** | **92.34** | 91.81 | 92.42 | **93.33** | 78.85 | **81.18** |
| **MATH-500** | 69.55 | **69.73** | 70.23 | **71.94** | 54.21 | **56.21** |
| **Comp-Math** | 75.20 | **76.16** | 73.65 | **75.09** | 57.03 | **58.52** |
| **ProcessBench** | 67.17 | **67.42** | 67.80 | **69.85** | 23.63 | **31.14** |
| **Average** | 74.73 | **75.24** | 74.31 | **76.01** | 56.50 | **58.97** |
| *Longbench Task* | | | | | | |
| **HotpotQA** | **60.81** | 58.61 | 54.43 | **61.56** | 46.35 | **55.19** |
| **Qasper** | 35.06 | **36.66** | **42.34** | 40.21 | 31.16 | **32.78** |
| **TriviaQA** | 79.96 | **83.25** | 75.79 | **84.40** | **46.34** | 41.82 |
| **NarrativeQA** | 18.87 | **19.19** | 16.54 | **25.43** | 20.95 | **25.45** |
| **2WikiMQA** | 70.81 | **71.27** | 72.43 | **73.46** | 62.32 | **68.23** |
| **MultiFieldQA** | 43.25 | **47.51** | 42.49 | **47.21** | 35.95 | **40.39** |
| **Average** | 51.46 | **52.75** | 50.67 | **55.38** | 40.51 | **43.98** |

with load balancing achieve lower loss and lower validation BPB, indicating improved learning in later stages.

### 5.3. Fine-Tuning Existing LLMs

To validate our approach on larger scales and state-of-the-art LLMs, we conduct fine-tuning experiments on Qwen3-4B, Qwen3-8B (Yang et al., 2025), and LLaMA3.1-8B (Dubey et al., 2024). Table 2 presents the results on reasoning and long-context benchmarks. Models fine-tuned with the auxiliary load balancing loss consistently outperform those trained with the base loss alone across task categories. Head importance after fine-tuning is shown in Appendix D.

Importantly, we observe that larger models benefit more substantially from balanced head utilization. This trend aligns with observations in prior MoE research (Shazeer et al., 2017; Fedus et al., 2021), where scaling to larger expert counts amplifies the importance of load balancing. The results demonstrate that our sink-aware training strategy effectively transfers to existing pretrained LLMs, providing a practical approach to enhance model performance through improved head utilization.

## 6. Further Analysis

### 6.1. A Mechanistic Hypothesis for Head Collapse

The analysis above shows that attention heads can be viewed as native experts, while sink induced gates act as routers. We now discuss why head collapse appears with different severity across attention mechanisms. Figure 9 tracks three training diagnostics: value drain, attention sink, and Head Load Imbalance CV. The results suggest a mechanistic hypothesis in which head collapse emerges after an operative gating mechanism has formed:

Value Drain → Attention Sink → Implicit Gating → Head Collapse

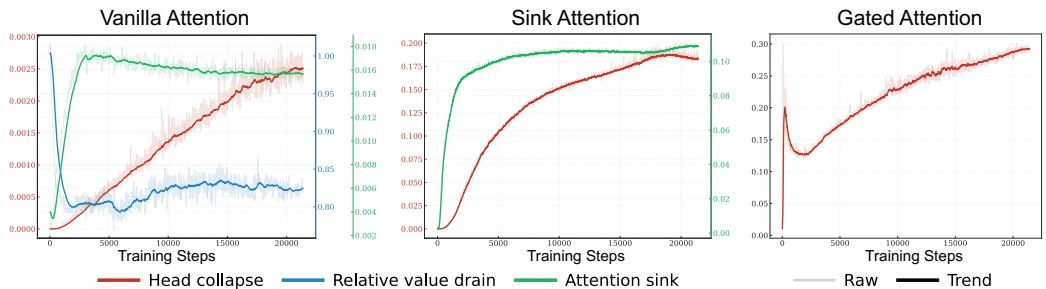

*Figure 9.* Training dynamics of value drain, attention sink, and head collapse. Value drain is measured by the ratio between the $\ell_2$ norm of the first token value vector and the average $\ell_2$ norm of other value vectors. Attention sink is measured by the mean attention weight assigned to the sink component. Head collapse is measured by Head Load Imbalance CV. The trajectories suggest that collapse becomes stronger once a gating mechanism is established.

This hypothesis explains the different collapse patterns observed in Section 4.1. In *Gated Attention*, the gating factor $G_t^{l,h}$ is explicit from the beginning of training. Therefore, head collapse can arise directly, similar to expert collapse in standard MoE models. This explains its early and severe imbalance.

In *Sink Attention*, the learnable *sink* has no associated value vector. Thus, the model does not need to first suppress the value of a real token. Once attention mass is assigned to the *sink*, the implicit gate $G_t^{l,h} = 1 - A_{t,\text{sink}}^{l,h}$ is formed, and head collapse can follow. This places *Sink Attention* between *Vanilla Attention* and *Gated Attention*.

In *Vanilla Attention*, the model must complete the full process. It first develops value drain, where the first token value vector becomes close to zero. This allows the first token to absorb redundant attention without changing the output. The resulting attention sink then induces the implicit gate in Equation (4). Only after this gate becomes effective does head collapse become pronounced. These phenomena need not occur in strictly separate stages. They can co-evolve during optimization. The key point is that a working gate is a prerequisite for collapse.

### 6.2. Limits of Single Sink Based Head Activation

Our analysis estimates head activation using the first token in *Vanilla Attention*, the *sink* parameter in *Sink Attention*, and the explicit gate in *Gated Attention*. Prior work shows that tokens other than the first token may also attract large attention weights (Yu et al., 2024). For example, punctuation tokens can become local attention sinks. However, they often aggregate contextual information from preceding tokens and thus cannot always serve as pure attention dumps (Chen et al., 2024). This differs from the first token, which has limited semantic content under causal masking and can more reliably act as a sink (Barbero et al., 2025).

However, we cannot rule out the existence of other low semantic tokens that absorb redundant attention. Let $\mathcal{S}^{l,h}$ denote the set of sink-like components. This set may include

the first token, early tokens, or context tokens with little semantic value. A more general implicit gate can be written as:

$$G_t^{l,h} = 1 - \sum_{j \in \mathcal{S}^{l,h}} A_{t,j}^{l,h},$$

where $A_{t,j}^{l,h}$ is the attention weight from token $t$ to component $j$. Equation (4) is a special case where $\mathcal{S}^{l,h}$ contains only one sink component.

This limitation also applies to *Sink Attention*. In some models, such as DeepSeek-V4 (DeepSeek-AI, 2026), the *sink* parameter does not absorb all redundant attention, and the first token can still receive substantial attention. In this case, $A_{t,\text{sink}}^{l,h}$ in Equation (4) should include both the attention weight on the *sink* parameter and the attention weight on the first token. More broadly, it should cover all components that absorb redundant attention while contributing little semantic value. A practical way to identify such components is to combine high attention weight with low value norm.

The same issue can arise in *Gated Attention*. Although it has an explicit gate, implicit gates may still form if some tokens absorb redundant attention. This suggests that head activation may be governed by multiple gates rather than a single factor. A complete characterization of these hidden sink sets and their interaction with explicit gates remains an important direction for future work.

## 7. Conclusion

This work presents a comprehensive analysis of the relationship among *Vanilla Attention*, *Sink Attention*, and *Gated Attention*. We clarify that attention sink functions as an implicit gating factor, thereby forming native MoE in attention layers. Furthermore, to address head collapse, we introduce a sink-aware auxiliary load balancing loss, which encourages more balanced utilization of attention heads. We conduct extensive experiments across both pre-training and fine-tuning scenarios, covering different attention mechanisms and model scales. The results demonstrate that our method significantly improves model performance.

## Acknowledgments

This work was supported in part by the National Key Research and Development Program under Grant 2024YFB4505004, in part by NSFC under Grant 62495102, Grant 92464104, and Grant 62341407, in part by Beijing Municipal Science and Technology Program under Grant Z241100004224015, in part by 111 Project under Grant B18001. We thank Yonggan Fu and Huaqing Zhang for their helpful discussions and valuable feedback.

## Impact Statement

This paper presents work whose goal is to advance the field of Machine Learning. There are many potential societal consequences of our work, none which we feel must be specifically highlighted here.

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

## A. Additional Validation on Value Drain Analysis

To further validate the analysis on value drain phenomenon described in Section 3.1, we conduct an experiment that explicitly zeros out the first token's value vector after generating the KV cache. If the model intentionally learns to nullify this value to absorb redundant attention, then this intervention should have minimal impact on performance.

We evaluate two settings: (1) zeroing the first token's value across all heads, and (2) selectively zeroing only for heads where the sink attention weight exceeds a threshold. Specifically, we define the head-wise sink attention ratio (Gu et al., 2024):

$$\alpha_{sink}^{l,h} = \frac{1}{T} \sum_{i=0}^{T-1} A_{i,sink}^{l,h}, \tag{9}$$

For the selective setting, we zero the first token's value only for heads with $\alpha_{\text{sink}}^{l,h} > \tau$, where $\tau = 0.75$.

Table 3 presents results on two models with *Vanilla Attention*. Zeroing across all heads causes only a negligible drop in performance. Moreover, selective zeroing for high-sink heads yields comparable or even slightly improved performance. This confirms that for heads exhibiting strong attention sink behavior, the first token's value does not carry useful semantic information and primarily serves to absorb redundant attention weight.

These results provide strong evidence that the value drain phenomenon is an intentional behavior learned by models with *Vanilla Attention*. The model deliberately suppresses the first token's value to enable its use as an attention sink without affecting the output computation.

*Table 3.* Performance comparison under different value zeroing settings. "None" denotes no intervention. "All" zeros the first token's value across all heads. "$\tau = 0.75$" selectively zeros for heads with sink attention ratio above 0.75.

| Model | Method | ARC | HellaSwag | MMLU | GSM8K | BBH | BoolQ | ToxiGen | TQA | HumanEval | AVG |
|---|---|---|---|---|---|---|---|---|---|---|---|
| Qwen3-8B | None | 68.96 | 74.97 | 75.42 | 87.64 | **79.31** | 86.73 | 45.96 | 50.24 | **64.02** | 70.36 |
| | All | **69.82** | 61.02 | 50.27 | 87.57 | 76.17 | 76.48 | 42.45 | 50.33 | 62.82 | 64.10 |
| | $\tau$=0.75 | 67.53 | **76.32** | **75.46** | **87.87** | 79.24 | **86.89** | **46.06** | **51.14** | 62.90 | **70.38** |
| Qwen3-14B | None | 71.49 | 78.84 | **79.05** | **82.41** | 41.88 | 89.30 | 47.13 | 56.06 | 56.1 | 66.91 |
| | All | **73.52** | 75.76 | 74.67 | 81.73 | **46.74** | 87.28 | 45.00 | 53.51 | 54.88 | 65.90 |
| | $\tau$=0.75 | 71.66 | **79.78** | **79.05** | 81.80 | 41.88 | **89.35** | **47.23** | **56.37** | **56.71** | **67.09** |

## B. Derivations for Implicit Gating in Vanilla Attention and Sink Attention

### B.1. Notation and Setup

We consider the $h$-th head in the $l$-th attention layer at time step $t$ with causal mask. For each real token index $j \in \{0, 1, \ldots, t\}$, we define the pre-softmax logit (i.e., scaled dot product) as

$$z_{t,j}^{l,h} \triangleq \frac{\mathbf{q}_t^{l,h} \mathbf{k}_j^{l,h\top}}{\sqrt{d_h}},$$

where $\mathbf{q}_t^{l,h} \in \mathbb{R}^{d_h}$ and $\mathbf{k}_j^{l,h} \in \mathbb{R}^{d_h}$ are the query and key vectors, and $d_h$ is the head dimension.

With these definitions, the output of the attention head at time step $t$ can be written as

$$O_t^{l,h} = \sum_j A_{t,j}^{l,h} \mathbf{v}_j^{l,h},$$

where $\mathbf{v}_j^{l,h} \in \mathbb{R}^{d_h}$ is the value vector, and $A_{t,j}^{l,h}$ is the attention weight.

For later derivations, we will use two standard operators: First, the sigmoid function is defined as $\sigma(u) \triangleq 1/(1 + \exp(-u))$. Second, for a set of logits $\{a_i\}$, we define the log-sum-exp operator as $\text{LSE}(\{a_i\}) \triangleq \log \sum_i \exp(a_i)$.

## B.2. Vanilla Attention: First Token as Attention Sink and Implicit Gating Factor

In *Vanilla Attention*, the attention weights over real tokens satisfy

$$A_{t,j}^{l,h} = \frac{\exp\left(z_{t,j}^{l,h}\right)}{\sum_{m=0}^{t} \exp\left(z_{t,m}^{l,h}\right)} \quad \text{for } j \in \{0,1,\ldots,t\}.$$

Following prior analyses of the attention sink and the value drain effect (Guo et al., 2024; Gu et al., 2024) in Section 3.1, we treat the first token as the sink component and use

$$\mathbf{v}_0^{l,h} \approx \mathbf{0},$$

meaning that allocating attention weight to $j = 0$ contributes negligibly to the output (Guo et al., 2024; Gu et al., 2024; Barbero et al., 2025). Under this approximation,

$$O_t^{l,h} = \sum_{j=0}^{t} A_{t,j}^{l,h} \mathbf{v}_j^{l,h} \approx \sum_{j=1}^{t} A_{t,j}^{l,h} \mathbf{v}_j^{l,h}. \tag{10}$$

To expose the implicit gating form, define the two denominators

$$D_t^{l,h} \triangleq \sum_{m=0}^{t} \exp\left(z_{t,m}^{l,h}\right), \qquad D_{t,\neg 0}^{l,h} \triangleq \sum_{m=1}^{t} \exp\left(z_{t,m}^{l,h}\right).$$

Next define the renormalized attention over non-sink tokens (the set $\{1,\ldots,t\}$):

$$\tilde{A}_{t,j}^{l,h} = \frac{\exp\left(z_{t,j}^{l,h}\right)}{D_{t,\neg 0}^{l,h}} \quad \text{for } j \in \{1,\ldots,t\}, \qquad \sum_{j=1}^{t} \tilde{A}_{t,j}^{l,h} = 1. \tag{11}$$

Now rewrite $A_{t,j}^{l,h}$ by inserting the factor $\frac{D_{t,\neg 0}^{l,h}}{D_{t,\neg 0}^{l,h}}$ into the denominator:

$$A_{t,j}^{l,h} = \frac{\exp\left(z_{t,j}^{l,h}\right)}{D_t^{l,h}} = \frac{\exp\left(z_{t,j}^{l,h}\right)}{D_{t,\neg 0}^{l,h}} \cdot \frac{D_{t,\neg 0}^{l,h}}{D_t^{l,h}} \quad \text{for } j \in \{1,\ldots,t\} \tag{12}$$

$$= \tilde{A}_{t,j}^{l,h} \cdot \frac{D_{t,\neg 0}^{l,h}}{D_t^{l,h}}. \tag{13}$$

The ratio $\frac{D_{t,\neg 0}^{l,h}}{D_t^{l,h}}$ is exactly the total weight assigned to non-sink tokens:

$$\frac{D_{t,\neg 0}^{l,h}}{D_t^{l,h}} = \frac{\sum_{m=1}^{t} \exp\left(z_{t,m}^{l,h}\right)}{\sum_{m=0}^{t} \exp\left(z_{t,m}^{l,h}\right)} = 1 - \frac{\exp\left(z_{t,0}^{l,h}\right)}{\sum_{m=0}^{t} \exp\left(z_{t,m}^{l,h}\right)} = 1 - A_{t,0}^{l,h}.$$

Substituting this identity and Equation (13) into Equation (10) yields the desired form:

$$O_t^{l,h} \approx \sum_{j=1}^{t} A_{t,j}^{l,h} \mathbf{v}_j^{l,h} = \left(1 - A_{t,0}^{l,h}\right) \cdot \sum_{j=1}^{t} \tilde{A}_{t,j}^{l,h} \mathbf{v}_j^{l,h}. \tag{14}$$

This matches the main text with sink $= 0$ for *Vanilla Attention* and $A_{t,\text{sink}}^{l,h} = A_{t,0}^{l,h}$.

**Sigmoid form in Vanilla Attention.** Let $\text{LSE}_{t,\neg 0}^{l,h}$ denote the log-sum-exp over non-sink logits:

$$\text{LSE}_{t,\neg 0}^{l,h} = \log \sum_{m=1}^{t} \exp\left(z_{t,m}^{l,h}\right) = \log D_{t,\neg 0}^{l,h},$$

Then the implicit gate $G_{\text{sink}}^{l,h} = 1 - A_{t,0}^{l,h}$ becomes

$$1 - A_{t,0}^{l,h} = \frac{D_{t,\neg 0}^{l,h}}{D_{t,\neg 0}^{l,h} + \exp\left(z_{t,0}^{l,h}\right)} \tag{15}$$

$$= \frac{1}{1 + \exp\left(z_{t,0}^{l,h} - \log D_{t,\neg 0}^{l,h}\right)} = \sigma\left(\text{LSE}_{t,\neg 0}^{l,h} - z_{t,0}^{l,h}\right), \tag{16}$$

where $z_{t,0}^{l,h} = \frac{\mathbf{q}_t^{l,h} \mathbf{k}_0^{l,h \top}}{\sqrt{d_h}}$. This is the sigmoid expression stated in the main text.

### B.3. Sink Attention: Parameter Sink as Implicit Gating Factor

In *Sink Attention*, an additional synthetic sink term is appended to the softmax denominator (Agarwal et al., 2025). We treat the sink logit as a learnable scalar parameter for each head and layer, denoted by $\mathbf{sink}^{l,h} \in \mathbb{R}$. The attention weights over real tokens are

$$A_{t,j}^{l,h} = \frac{\exp\left(z_{t,j}^{l,h}\right)}{\sum_{m=0}^{t} \exp\left(z_{t,m}^{l,h}\right) + \exp\left(\mathbf{sink}^{l,h}\right)} \quad \text{for } j \in \{0, 1, \ldots, t\},$$

and the attention weight on the synthetic sink is

$$A_{t,\text{sink}}^{l,h} = \frac{\exp\left(\mathbf{sink}^{l,h}\right)}{\sum_{m=0}^{t} \exp\left(z_{t,m}^{l,h}\right) + \exp\left(\mathbf{sink}^{l,h}\right)}.$$

The key design is that the sink contributes zero to the output. This can be implemented by assigning it a zero value vector, $\mathbf{v}_{\text{sink}}^{l,h} = \mathbf{0}$, or by treating it as a denominator-only term with no associated value (Agarwal et al., 2025). In either view,

$$A_{t,\text{sink}}^{l,h} \mathbf{v}_{\text{sink}}^{l,h} = \mathbf{0}.$$

Therefore the head output reduces to the sum over real tokens:

$$O_t^{l,h} = \sum_{j=0}^{t} A_{t,j}^{l,h} \mathbf{v}_j^{l,h}. \tag{17}$$

Define the token-only denominator and the full denominator as:

$$D_{t,\text{tok}}^{l,h} \triangleq \sum_{m=0}^{t} \exp\left(z_{t,m}^{l,h}\right), \qquad D_{t,\text{all}}^{l,h} \triangleq D_{t,\text{tok}}^{l,h} + \exp\left(\mathbf{sink}^{l,h}\right).$$

Also define the renormalized attention over real tokens:

$$\tilde{A}_{t,j}^{l,h} = \frac{\exp\left(z_{t,j}^{l,h}\right)}{D_{t,\text{tok}}^{l,h}} \quad \text{for } j \in \{0, 1, \ldots, t\}, \qquad \sum_{j=0}^{t} \tilde{A}_{t,j}^{l,h} = 1. \tag{18}$$

Now apply the same denominator factorization:

$$A_{t,j}^{l,h} = \frac{\exp\left(z_{t,j}^{l,h}\right)}{D_{t,\text{all}}^{l,h}} = \frac{\exp\left(z_{t,j}^{l,h}\right)}{D_{t,\text{tok}}^{l,h}} \cdot \frac{D_{t,\text{tok}}^{l,h}}{D_{t,\text{all}}^{l,h}} \quad \text{for } j \in \{0, \ldots, t\} \tag{19}$$

$$= \tilde{A}_{t,j}^{l,h} \cdot \frac{D_{t,\text{tok}}^{l,h}}{D_{t,\text{all}}^{l,h}}. \tag{20}$$

The ratio is again the complement of the sink weight:

$$\frac{D_{t,\text{tok}}^{l,h}}{D_{t,\text{all}}^{l,h}} = 1 - \frac{\exp\left(\mathbf{sink}^{l,h}\right)}{D_{t,\text{tok}}^{l,h} + \exp\left(\mathbf{sink}^{l,h}\right)} = 1 - A_{t,\text{sink}}^{l,h}.$$

Substituting Equation (20) into Equation (17) yields

$$O_t^{l,h} = \left(1 - A_{t,\text{sink}}^{l,h}\right) \cdot \sum_{j=0}^{t} \tilde{A}_{t,j}^{l,h} \, \mathbf{v}_j^{l,h}, \tag{21}$$

which aligns the main text with $A_{t,\text{sink}}^{l,h} = \frac{\exp(\mathbf{sink}^{l,h})}{\sum_m \exp(z_{t,m}^{l,h}) + \exp(\mathbf{sink}^{l,h})}$.

**Sigmoid form in Sink Attention.** Let the token-only log-sum-exp be

$$\text{LSE}_{t,\text{tok}}^{l,h} = \log \sum_{m=0}^{t} \exp\left(z_{t,m}^{l,h}\right) = \log D_{t,\text{tok}}^{l,h},$$

Then, the implicit gate becomes:

$$1 - A_{t,\text{sink}}^{l,h} = \frac{D_{t,\text{tok}}^{l,h}}{D_{t,\text{tok}}^{l,h} + \exp\left(\mathbf{sink}^{l,h}\right)} \tag{22}$$

$$= \frac{1}{1 + \exp\left(\mathbf{sink}^{l,h} - \log D_{t,\text{tok}}^{l,h}\right)} = \sigma\left(\text{LSE}_{t,\text{tok}}^{l,h} - \mathbf{sink}^{l,h}\right). \tag{23}$$

This is the sigmoid expression stated in the main text.

### B.4. Connection to the Implicit Gate Used in the Main Text

Both derivations yield the same implicit gating definition:

$$G_{\text{sink}}^{l,h} = 1 - A_{t,\text{sink}}^{l,h},$$

where $A_{t,\text{sink}}^{l,h} = A_{t,0}^{l,h}$ for *Vanilla Attention* (using $\mathbf{v}_0^{l,h} \approx \mathbf{0}$) and $A_{t,\text{sink}}^{l,h}$ is the synthetic sink weight for *Sink Attention* (using $\mathbf{v}_{\text{sink}}^{l,h} = \mathbf{0}$) (Guo et al., 2024; Gu et al., 2024; Barbero et al., 2025; Agarwal et al., 2025).

## C. Additional Results on Head Collapse

This appendix provides comprehensive evidence of head collapse across a diverse range of large language models. We extend the analysis in Section 4.1 by examining head importance patterns in models spanning different architectures, families, and parameter scales.

We compute the head importance scores defined in Equation (5) by running inference on 500 samples from the GSM8K dataset. For each model, we collect the gating factors across all tokens and aggregate them to obtain per-head importance scores. Most models use *Vanilla Attention* and *Sink Attention* where we apply the implicit gating factor from Equation (4). For Qwen3-Next-80B-A3B, which employs head-specific element-wise gating factors with heterogeneous layer types (linear layers and full attention layers with different head counts), we compute importance scores by first averaging the gating factors across dimensions within each head, then applying Equation (5).

**Results on GPT-OSS and LLaMA Families.** Figure 10 presents head importance heatmaps for models from the GPT-OSS (Agarwal et al., 2025) and LLaMA (Dubey et al., 2024) families, including GPT-OSS-20B, GPT-OSS-120B, LLaMA-2-7B, LLaMA-2-13B, LLaMA-2-70B, and LLaMA-3.3-70B. Across all models, we observe pronounced head collapse with a small subset of heads consistently showing high importance scores while many others remain largely inactive. Notably,

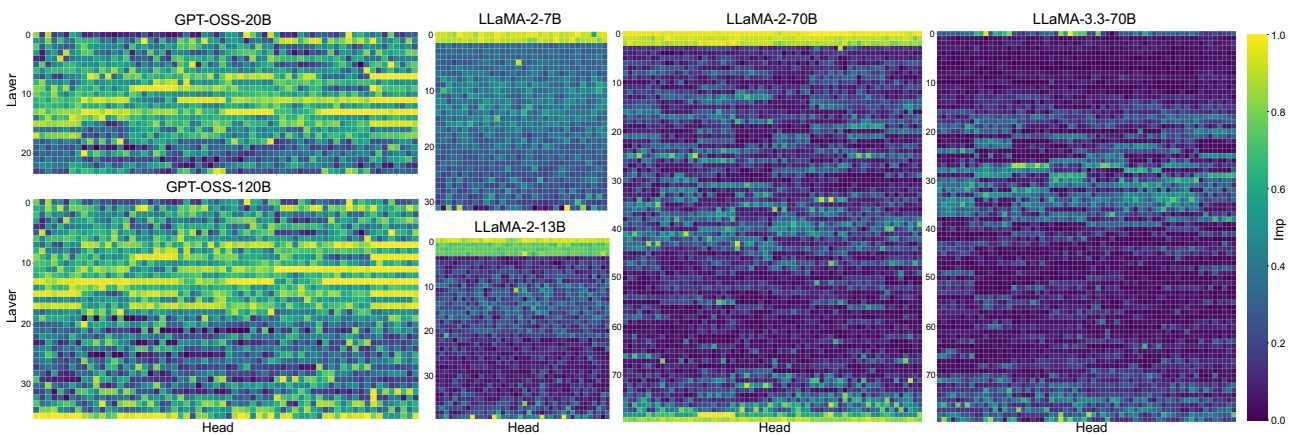

*Figure 10.* Head importance scores across the GPT-OSS, LLaMA-2, and LLaMA-3 families.

the imbalance becomes more severe as model size increases. Larger models exhibit a more extreme disparity between active and dormant heads, suggesting that head collapse intensifies with scale.

**Results on Qwen Family.** Figure 11 shows head importance distributions for the Qwen family (Yang et al., 2025), including Qwen2.5-0.5B, Qwen2.5-3B, Qwen3-4B, Qwen2.5-7B, Qwen3-8B, Qwen3-14B, Qwen3-30B-A3B, Qwen2.5-32B, Qwen3-32B, and Qwen2.5-72B. The pattern of head collapse is consistent across the entire Qwen family. As with the GPT-OSS and LLaMA models, head utilization becomes increasingly imbalanced with growing model capacity.

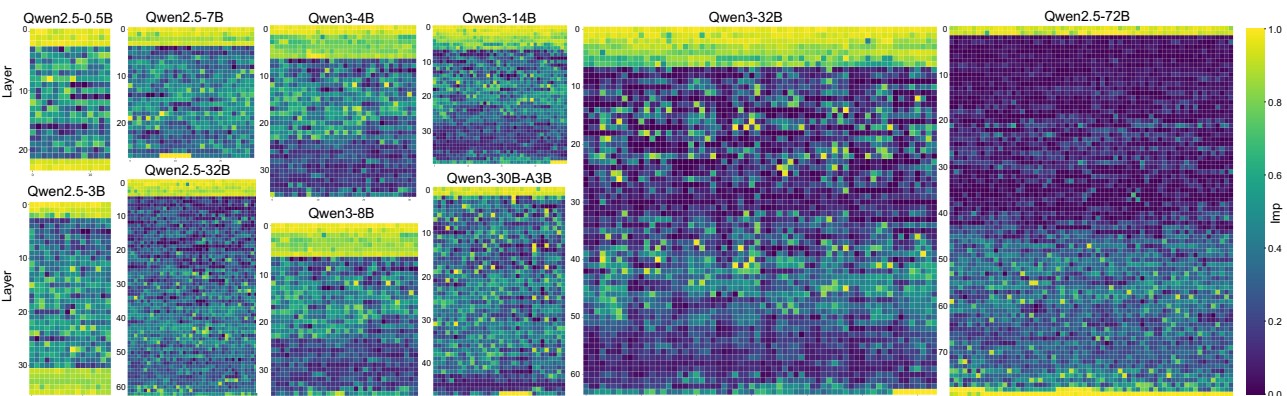

*Figure 11.* Head importance scores across the Qwen2.5 and Qwen3 families. Larger models demonstrate more severe head collapse with lower overall head utilization.

**Key Observations.** The results across all three model families reveal a clear trend: head collapse intensifies with model scale. Models with more parameters and more attention heads paradoxically exhibit lower head utilization rates. This suggests that simply scaling up the number of heads does not guarantee better capacity utilization. Instead, as models grow larger, the optimization process increasingly concentrates the workload on a fixed subset of heads, leaving many heads underutilized. This finding motivates our sink-aware training approach to promote more balanced head utilization.

## D. Head Pinning Effect in Fine-Tuning Existing LLMs

When fine-tuning existing LLMs with the auxiliary load balancing loss defined in Equation (7), we observe an unexpected phenomenon that we term the *head pinning effect*. Figure 12 illustrates this behavior on Qwen3-8B. The middle panel shows that after fine-tuning with Equation (7), nearly all heads exhibit uniformly high importance scores. While this appears to satisfy the balancing objective, it does not achieve our intended goal of balanced head utilization.

This phenomenon arises because collapsed heads have become dominant and essential for generation during pretraining.

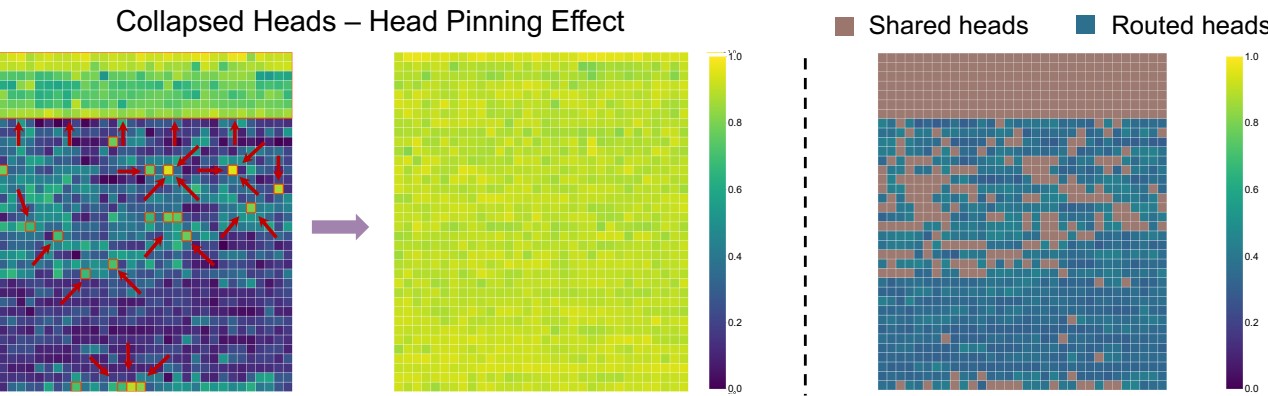

*Figure 12.* Head importance scores $\mathrm{Imp}^{l,h}$ for Qwen3-8B across three conditions: before fine-tuning (left), after one epoch of fine-tuning with Equation (7) (middle), and after one epoch of fine-tuning with Equation (8) (right).

These heads carry critical learned representations, and reducing their influence causes performance degradation. Consequently, their gating factors resist change during fine-tuning. When the auxiliary loss pushes for uniformity, the remaining heads adapt by increasing their own gating factors to align with the dominant heads.

To address this issue, we adopt a strategy inspired by the shared expert and routed expert design in DeepSeek-MoE (Dai et al., 2024). We identify the top $m$ most important heads in each layer as *shared heads*, which remain active and preserve their learned behaviors. The remaining heads serve as *routed heads*, for which we apply the auxiliary load balancing loss. This approach is formalized in Equation (8), where only the routed heads are subject to the balancing objective.

The right panel of Figure 12 demonstrates the effectiveness of this refined approach. Fine-tuning with Equation (8) produces a more balanced distribution of importance scores among routed heads, while preserving the high activation of shared heads. This configuration achieves our goal of improved head utilization without disrupting the critical representations learned during pretraining.

