# OpenReview forum: "Attention Sink Forges Native MoE in Attention Layers: Sink-Aware Training to Address Head Collapse"
_ICML.cc/2026/Conference — ICML 2026 regular_

### Official Review · Reviewer_Rvs6 · 2026-02-25

**Soundness:** 3
**Presentation:** 3
**Significance:** 3
**Originality:** 3
**Overall Recommendation:** 5
**Confidence:** 4

**Summary:**

This work interprets the attention weights assigned to sink tokens in both vanilla attention and sink attention as implicit gating factors and draws a comparison with the explicit gating factors used in gated attention mechanisms. This perspective provides a unified view of these three attention paradigms. The authors further relate implicit and explicit gating to the Mixture-of-Experts (MoE) framework, suggesting that attention sinks can induce an MoE-like structure. Based on this view, they quantify head importance using both implicit and explicit gating factors, which serve as indicators of each head’s contribution to generation. Empirical results show that, across the three attention mechanisms, many heads receive low importance scores, a phenomenon referred to as head collapse. To mitigate this issue, the authors introduce a sink-aware load balancing loss that encourages more uniform importance scores across heads within each layer. Experiments indicate that this loss, together with an additional term designed to address the head pinning effect, can improve performance in both training-from-scratch and fine-tuning settings.

**Compliance With Llm Reviewing Policy:**

Affirmed.

**Final Justification:**

After reviewing the rebuttal, I am raising my score to a 5. The authors successfully clarified that the gating score is treated as a continuous importance measure, which convincingly resolved my initial questions. Additionally, they provided a strong analysis validating the gating-based importance metric, strengthening the methodological soundness of the work. Furthermore, the newly included discussions regarding head utilization and redistribution are insightful and add valuable depth to the paper. Since my primary concerns have been effectively addressed, my final evaluation is positive.

**Key Questions For Authors:**

- Do the models trained with and without load balancing loss present attention sink phenomenon?

- As shown in [1] and [2], there may be multiple sink tokens within a sequence. Was this possibility taken into account when designing the load balancing loss?

**Reference**

[1] Massive activations in large language models.

[2] Unveiling and Harnessing Hidden Attention Sinks: Enhancing Large Language Models without Training through Attention Calibration.

**Limitations:**

Yes

**Strengths And Weaknesses:**

**Strengths:**
- This work creatively treats the attention weights assigned to sink tokens as gating factors and establishes a connection between head collapse and expert collapse in the Mixture-of-Experts (MoE) literature. From this perspective, the proposed load balancing loss becomes both intuitive and well-motivated.

-  The paper provides several novel empirical observations. For example, in Figure 3, the authors show that the diversity of QK vectors may be constrained in vanilla attention, extending the conclusions reported in [1].

- Extensive experiments are conducted across various model families and widely used benchmarks, which strengthens the empirical credibility of the paper.

- The authors leverage log-sum-exp values to make the computation of gating factors compatible with FlashAttention modules, introducing less than 2% additional training latency. This design demonstrates strong practical awareness.

**Weaknesses:**

- In Equation (4), the implicit gating factor is defined as $G = 1 - A_{sink}$ (using simplified notation). Interpreting $G$ as a gating factor implies that when $G$ is close to zero, most attention mass is assigned to the first token and the head is considered inactive with a nearly zero value vector, whereas when $G$ is close to one, the head is activated. However, the paper does not provide a clear criterion for how small $G$ must be for a head to be regarded as inactive. Following the convention in [1], when the sequence length $T=64$, $G\leq 0.7$ already corresponds to attention sink behavior. In this regime, $G$ does not need to be close to zero, and head collapse would not necessarily occur if one focuses solely on the magnitude of $G$. Therefore, the definition and threshold for collapse remain somewhat ambiguous.

- In Section 4, the authors use the average gating factor to measure head importance. However, the effectiveness and robustness of this metric are not theoretically justified or empirically validated. In fact, [4] proposes several alternative metrics and shows (Table 2) that different models may require different criteria to determine whether a head is inactive. While existing literature suggests that a head exhibiting attention sink patterns together with small value-vector norms can be roughly considered dormant, the paper would benefit from a more detailed discussion or additional validation supporting the proposed importance metric.

-  As shown in [1], attention sink behavior appears in nearly all popular LLMs, with most attention heads exhibiting this pattern. [2] demonstrates that sink tokens must be deliberately retained to maintain model stability at long context lengths. Similarly, [3] argues that attention sink is a learned mechanism that helps prevent excessive token mixing. Therefore, it remains unclear in the current literature whether heads that primarily attend to the first token represent parameter redundancy or serve a necessary functional role that is learned automatically by models. Although the authors cite works suggesting that some heads contribute little to final performance, their functional role during pre-training is not examined. Moreover, the reported performance improvements are relatively modest (less than 2 points), which does not fully establish that these heads are collapsed and redundant heads are better utilized through the proposed load balancing loss. In Figure 6, while load balancing appears improved, this does not necessarily imply better overall head utilization—it could instead reflect a redistribution or averaging of computation rather than genuinely enhanced functional contribution.

**Reference**

[1]  When attention sink emerges in language models: An empirical view.

[2] Efficient streaming language models with attention sinks.

[3] Why do LLMs attend to the first token?

[4] Identifying and Evaluating Inactive Heads in Pretrained LLMs.

---

> ### Author Rebuttal · Authors · 2026-03-30
>
> We sincerely thank Reviewer Rvs6 for the careful reading and valuable feedback. Below we address the raised concerns in detail.
>
> **Please see the newly added experimental results at [https://anonymous.4open.science/api/repo/ICML-2026-395/file/ICML_2026_response.pdf](https://anonymous.4open.science/api/repo/ICML-2026-395/file/ICML_2026_response.pdf).**
>
> ---
>
> **W1: Threshold for defining head collapse.**
>
> Our work focuses on the *degree* of activation rather than a binary status. A threshold-based classification (active vs. inactive), as in [1], is indeed feasible. However, the gating factor (G) mirrors the continuous MoE gating weight and quantifies *how strongly* a head contributes. Since prior works [1, 2] have already studied binary activation, **our contribution lies in treating (G) as a continuous importance measure**, which more naturally extends the MoE analogy.
>
> ---
>
> **W2: Validity of the gating-based importance metric.**
>
> **Our metric is both effective and consistent with alternative measures.** We provide two pieces of evidence.
>
> * **Effectiveness.** [2] shows that zeroing out over 25% of heads selected by the attention-sink metric preserves model accuracy, confirming that this metric reliably identifies functionally inactive heads.
> * **Consistency.** Figure 8 (attached pdf) compares our gating-based metric with the per-head output $\ell_2$ norm from [1]. **The two metrics yield highly consistent head-importance patterns across all three attention variants.** This is expected: attention sink and value drain jointly suppress output magnitude, so stronger sink effects naturally yield smaller output norms. While individual metrics may disagree on borderline cases, they identify largely the same set of inactive heads, supporting the robustness of our metric.
>
> ---
>
> **W3: Functional role of sink heads, performance, and head utilization.**
>
> We address this concern in three parts.
>
> * **Are sink heads truly redundant?** [2] shows that removing sink-dominated heads causes minimal accuracy loss, indicating limited functional contribution. This is consistent with [3]: attention sink routes surplus attention mass to the semantically impoverished first token precisely to prevent excessive token mixing.
>
> * **Magnitude of improvements.** Fine-tuning on Qwen3-8B and LLaMA3.1-8B (Table 2 of the original paper) shows gains of up to **4.71 points**. Furthermore, Table 2 (attached pdf) reports results for a wider 1B model (16 layers, 32 heads per layer) trained from scratch, where the auxiliary loss yields an average accuracy gain of **2.75 points**. This suggests the benefit scales with the number of heads per layer.
>
> * **Head utilization vs. redistribution.** Load balancing does not aim for uniform activation at every decoding step. **The goal is task-adaptive specialization, analogous to expert specialization in MoE.** Figure 6 (attached pdf) shows that after applying the auxiliary loss, heads exhibit distinct task preferences. Figure 7 (attached pdf) further shows that head activations become more evenly distributed along the sequence dimension, with greater differentiation across heads. These results confirm genuine functional specialization rather than mere redistribution.
>
> ---
>
> **Q1: Do trained models still exhibit attention sinks?**
>
> **Yes, attention sink behavior persists in both settings, but the auxiliary loss makes the sink pattern more evenly distributed across heads.** Figure 9 (attached pdf) shows attention sink heat maps for 0.6B and 2B base-trained models. The auxiliary loss does not eliminate sinks; it redistributes them more uniformly. For fine-tuning, Appendix D of the original paper shows a similar trend.
>
> ---
>
> **Q2: Multiple sink tokens within a sequence.**
>
> **We argue that only the first token functions as a true gating sink.** While punctuation tokens also attract elevated attention, they differ from the first token on two grounds.
>
> * [4] notes that punctuation tokens aggregate rich contextual information from preceding tokens and therefore cannot serve as pure attention dumps.
> * Figure 10 (attached pdf) plots token-wise value $\ell_2$ norms in Qwen2.5-7B, Qwen3-8B, and LLaMA-3.1-8B. Although non-initial punctuation tokens attract attention, **their value norms do not exhibit the pronounced drain observed at the first token**, indicating that they still carry non-trivial semantic content.
>
> This aligns with [3]: due to the causal mask, only the first token has minimal semantic content and can reliably act as a gating sink, which is why our load balancing loss targets it.
>
> ---
>
> **References**
>
> [1] Identifying and Evaluating Inactive Heads in Pretrained LLMs
>
> [2] Garbage Attention in Large Language Models
>
> [3] Why do LLMs attend to the first token?
>
> [4] SepLLM: Accelerate Large Language Models by Compressing One Segment into One Separator
>
> ---
>
> **We thank the reviewer's insightful suggestions and will incorporate these discussions into the revised version.**

---

> > ### Author Rebuttal · Reviewer_Rvs6 · 2026-04-03
> >
> > Thanks for your response. My concerns have been addressed, and I would like to increase my score.

---

> > > ### Author Response · Authors · 2026-04-03
> > >
> > > We sincerely thank you for your encouraging response and for bringing up such a stimulating discussion.
> > >
> > > The discussion on *multiple sink tokens within a sequence* was particularly insightful, as it helped us sharpen the logical chain underlying our framework and meaningfully strengthened the paper. We are grateful for this valuable exchange. If the paper is accepted, **we will incorporate these discussions into the final version.**
> > >
> > > **Thank you again for your time and thoughtful engagement throughout this process!**

---

### Official Review · Reviewer_K9qU · 2026-03-02

**Soundness:** 2
**Presentation:** 4
**Significance:** 3
**Originality:** 3
**Overall Recommendation:** 4
**Confidence:** 4

**Summary:**

This paper demonstrates that the attention sink phenomenon in Large Language Models (LLMs) naturally forges an implicit Mixture-of-Experts (MoE) structure within attention layers, which explains the cause of head collapse phenomenon observed in prior works. To mitigate head collapse, the authors propose sink-aware training method, which uses a load balancing auxillary loss function. Extensive experiments across Vanilla, Sink, and Gated Attention mechanisms show that addressing head collapse significantly improves model performance and representational capacity, particularly in long-context scenarios.

**Compliance With Llm Reviewing Policy:**

Affirmed.

**Final Justification:**

- **Visual Evidence**

The authors added more visualization of head importance heatmaps before and after applying the auxillary load balancing loss at my request (Figure 3 in their attachment file). The added visualization clearly shows that proposed load balancing loss alleviates attention sink phenomenon, which fully resolves my concern.

- **Causal Validation**

The authors conducted the anti-intuition experiment at my request （Table 1 & Figure 1）, which isolates the effect of the newly introduced load balancing loss, showing that positive $\lambda$ encourages load balancing while negative $\lambda$ causes more severe head collapsing. This also resolves my concern.

- **Why Is Head Collapse More Severe in Gated Attention?**

In response to this question, the authors propose an interesting hypothetic explanation that three attention variants differ in steps required to traverse the pipeline Value Drain → Attention Sink → Implicit Gating → Head Collapse. This is a new perspective and I appreciate their effort in proposing such hypothesis.

Overall, my major concerns have been resolved, so I raised my score to 4. Because the "value drain → … → head collapse" hypothesis is hard to validate and I conjecture there might be more mechanisms involved, I did not raise my score to 5.

**Key Questions For Authors:**

In Figure 4, the Gated Attention model (Qwen3-Next-80B-A3B) exhibits a very stark imbalance pattern where a small subset of heads has high importance while most are nearly zero. However, this disparity appears visually much less pronounced in the Vanilla and Sink Attention models. Could the authors explain why the visual evidence for collapse seems so much more severe in the gated variant compared to the others?

**Limitations:**

See weaknesses and questions.

**Strengths And Weaknesses:**

### Strengths

1. **Novelty**. The paper provides a quite novel perspective by analyzing the attention sink phenomenon through the lens of gating mechanisms, offering both theoretical and empirical explanations for the "head collapse" phenomenon and its negative impacts.
2. **Effectiveness**. The proposed method for eliminating attention sinks is technically sound, and the experimental results demonstrate consistent performance improvements across various benchmarks.
3. **Presentation**. The manuscript is clearly written, well-structured, and easy to follow, making the complex relationship between attention variants and MoE structures accessible.

### Weaknesses

1. **Visual Evidence**. To better illustrate the efficacy of the proposed solution, I suggest including a side-by-side visualization of head importance heatmaps before and after applying the auxiliary load balancing loss. This would provide a direct visual confirmation of how the "collapsed" heads are reactivated.
2. **Causal Validation**. The causal link between head imbalance and performance degradation requires more rigorous verification. Correlation does not imply causation. Notably, Figure 5 shows that Vanilla Attention has a significantly lower imbalance CV compared to Sink and Gated Attention ($10^{-4}$ vs. $10^{-1}$), yet its performance is inferior. To strengthen the argument, I recommend an "anti-intuition" experiment: modify the auxiliary loss to deliberately increase imbalance and observe if performance drops significantly.

---

> ### Author Rebuttal · Authors · 2026-03-30
>
> We sincerely thank Reviewer K9qU for the thorough and constructive feedback. Below we address the raised concern in detail.
>
> **Please see the newly added experimental results at [https://anonymous.4open.science/api/repo/ICML-2026-395/file/ICML_2026_response.pdf](https://anonymous.4open.science/api/repo/ICML-2026-395/file/ICML_2026_response.pdf).**
>
> ---
>
> **W1: Visual Evidence**
>
> We appreciate this suggestion and note that the original paper already contains relevant visualizations. In Appendix D, head importance heatmaps for fine-tuned models before and after applying the auxiliary load balancing loss show that routed head activation becomes substantially more balanced. In Figure 6, per-head activations across datasets within a single layer demonstrate that the auxiliary loss not only improves load balancing but also induces clearer task-specific head preferences, resembling the functional specialization observed in MoE models.
>
> **In response to this suggestion, we additionally provide Figure 3 (attached pdf), which presents comprehensive side-by-side importance heatmaps for 2B models trained from scratch under all three attention mechanisms, with and without the auxiliary loss. These results directly confirm that collapsed heads are successfully reactivated.**
>
> ---
>
> **W2: Causal Validation**
>
> We thank the reviewer for this insightful suggestion and **have conducted the requested anti-intuition experiment by setting the auxiliary loss coefficient λ to negative values, thereby deliberately promoting head imbalance**.
>
> * On 0.6B models spanning all three attention variants (Table 1, attached pdf), a negative λ causes a substantial drop in average accuracy and a marked increase in Validation BPB. **This demonstrates that artificially induced head imbalance causally degrades performance.**
> * A **sensitivity analysis on 1B models** (Figure 1, attached pdf) further shows that positive λ (encouraging balance) consistently improves performance, while negative λ (promoting imbalance) causes a sharp decline. **These bidirectional results establish a causal link between head imbalance and performance degradation.**
>
> Regarding the apparent counter-example in Figure 5 of the original paper (Vanilla Attention has lower CV yet inferior performance): this is explained by the mechanistic differences discussed in Q1 below. Vanilla Attention's lower CV reflects its multi-stage, implicit gating pathway rather than superior head utilization; its inferior overall performance instead stems from the constrained query-key geometry inherent to the standard softmax attention formulation, as analyzed in detail in Section 3.2 of the paper.
>
> ---
>
> **Q1: Why Is Head Collapse More Severe in Gated Attention?**
>
> This is an insightful question. **We propose a mechanistic hypothesis, supported by training dynamics in Figure 2 (attached pdf), which tracks value drain, attention sink, and Head Load Imbalance CV throughout training: head collapse emerges only after a gating mechanism is established, and the three variants differ in the number of prerequisite steps required.**
>
> The underlying pipeline is:
> [Value Drain → Attention Sink → Implicit Gating → Head Collapse]
>
> * **Gated Attention** possesses an *explicit* gating factor, so severe head collapse arises directly and very early in training, analogous to expert collapse in standard MoE models.
> * **Sink Attention** has a structurally zero-valued sink parameter, which eliminates the need for value drain. Collapse therefore follows once the attention sink pattern is learned, placing it as an intermediate case.
> * **Vanilla Attention** must traverse the full pipeline: the model first develops *value drain* (the first-token value vector collapses toward zero), which enables *attention sink*, which in turn establishes an implicit gate — only then does head collapse emerge. This multi-stage dependency slows the process considerably.
>
> **This explains why, under an identical training budget, Gated Attention exhibits the most pronounced collapse, Sink Attention is intermediate, and Vanilla Attention shows the least — a pattern consistent with both our training-from-scratch experiments and the large-scale pretrained models shown in Figure 5 of the original paper.** We emphasize that these phenomena can co-evolve rather than occurring in strictly sequential stages; the key point is that an operative gating mechanism is a *prerequisite* for head collapse.
>
> ---
>
> **We thank the reviewer for the valuable suggestions and will incorporate these discussions in the revised version.**

---

> > ### Author Rebuttal · Reviewer_K9qU · 2026-04-01
> >
> > Thank you for your efforts! I have carefully read the whole response and I find the explanation of the reason why vanilla attention has smaller CV value especially interesting. I think thoroughly understanding the reason may require more investigations into it, for example: after the pipeline is fully traversed and the head collapse values of all three structures reach the plateau, would the head collapse still be most severe for gated attention? But I understand that this requires much more effort and is not a problem that can be solved in such a short period of rebuttal. (Also this may not be that relevant to the topic of this paper)
> >
> > Finally, based on the authors response, I think the paper is worthy of publication and I would raise my score to 4 (weak accept).

---

> > > ### Author Response · Authors · 2026-04-02
> > >
> > > We sincerely thank you for the kind recognition and for raising such an insightful question throughout the review process.
> > >
> > > The mechanistic pipeline *Value Drain → Attention Sink → Implicit Gating → Head Collapse* is indeed a fascinating research direction. This question — whether gated attention would still suffer the most severe collapse after the full pipeline is traversed and all three variants reach a stable plateau — is genuinely thought-provoking and deserves further study.
> > >
> > > As you noted, answering this question well would require substantially longer training runs, more training data, and additional experiments, which unfortunately cannot be completed within the rebuttal period. We will continue to investigate this mechanism in future work and aim to provide a more detailed explanation.
> > > If the paper is accepted, **we will incorporate the relevant discussion into the final version**.
> > >
> > > We are truly excited by the discussion during the rebuttal process! The exchange has significantly deepened the theoretical grounding of this work, and the mechanistic insights raised here will make the paper substantially more complete. **Thank you again for your efforts and the thoughtful feedback!**

---

### Official Review · Reviewer_QbW5 · 2026-03-10

**Soundness:** 3
**Presentation:** 3
**Significance:** 3
**Originality:** 3
**Overall Recommendation:** 5
**Confidence:** 4

**Summary:**

This paper investigates the common attention sink phenomenon within attention mechanisms. To address this, the authors provide a unified modeling approach for three distinct attention architectures using a framework similar to Mixture of Experts (MoE). Through Principal Component Analysis, authors reveal that attention sinks lead to a restricted query-key space. Drawing inspiration from the critical load-balancing challenges in MoE training, authors propose an auxiliary loss designed to promote the balanced utilization of attention heads. Experimental results demonstrate that incorporating this auxiliary loss yields consistent performance gains across various model sizes and attention architectures, particularly in reasoning and long-context tasks.

**Compliance With Llm Reviewing Policy:**

Affirmed.

**Ethical Review Flag:**

Flag this paper for an ethics review.

**Final Justification:**

My concerns have been addressed, and I now believe the paper is solid both theoretically and empirically. Therefore, I have increased my score.

**Key Questions For Authors:**

I am curious about whether introduction of the auxiliary loss might interfere with the beneficial properties of attention mechanisms. For instance, after applying this loss, are we still able to effectively identify retrieval heads?

**Limitations:**

yes

**Strengths And Weaknesses:**

Strength:
1. The authors successfully unify different attention mechanisms through an MoE-style framework. This abstraction provides a natural motivation for introducing an Auxiliary Load Balancing Loss and allows for the seamless integration of techniques such as the "shared experts" mechanism popularized by DeepSeek-MoE.
2. The improvements are consistent across various model sizes and architectural variants. The proposed auxiliary loss demonstrates reliable performance gains in both reasoning and long-context benchmarks, reinforcing the generality of the method.
3. The proposed approach is well-aligned with real-world deployment constraints. Crucially, it maintains compatibility with Flash Attention, ensuring that the introduction of the auxiliary loss does not impose significant training overhead.

Weakness:
1. While the individual components are logical, the transition between sections is suboptimal. The link between the MoE-style framework in Section 3.1 and the PCA-based analysis of attention sinks in Section 3.2 is weak, as the MoE-style framework is essentially dropped during the analysis. Furthermore, the experimental section fails to explicitly demonstrate whether the proposed auxiliary loss effectively alleviates the identified restriction in the $QK$ space.
2. From an interpretability standpoint, the contribution is modest. While the MoE-style framework serves an engineering purpose, it lacks conceptual depth. The PCA-based analysis in Section 3.2 largely overlaps with existing literature regarding effective rank and intrinsic dimensionality, offering little in terms of new discovery. Moreover, the authors establish a correlation between attention sinks and a restricted $QK$ space, but they fail to provide definitive evidence that this restriction is the causal factor for performance degradation.

---

> ### Author Rebuttal · Authors · 2026-03-30
>
> We sincerely thank Reviewer QbW5 for the thorough and constructive review. Below we address the raised concerns in detail.
>
> **Please see the newly added experimental results at [https://anonymous.4open.science/api/repo/ICML-2026-395/file/ICML_2026_response.pdf](https://anonymous.4open.science/api/repo/ICML-2026-395/file/ICML_2026_response.pdf).**
>
> ---
>
> **W1: Weak link between Sections 3.1 and 3.2; auxiliary loss not shown to alleviate QK restriction.**
>
> We acknowledge this gap and apologize for the lack of clarity, and we now provide direct experimental evidence on QK restriction.
>
> * **On the connection between sections.** Section 3.2 is not directly about head collapse. It uses the unified framework to analyze all three mechanisms, offering a hypothesis for why vanilla attention underperforms: it relies on the sink mechanism for implicit head gating, which structurally restricts its QK space.
>
> * **On whether the auxiliary loss alleviates QK restriction.** Figure 4 (attached pdf) compares QK PCA distributions for 2B pretrained models and Qwen3-8B fine-tuned models, with and without the auxiliary loss. **The results show that load balancing partially alleviates the restricted QK pattern, with activating tokens exhibiting a broader QK distribution.**
>
> * **On why alleviation is only partial.** The auxiliary loss is not designed to eliminate attention sinks, but to leverage the gating structure they induce for balanced head utilization. The restriction in vanilla attention is structural: activating tokens move farther from key₀ while inactive ones remain near it. Our loss encourages more tokens to engage each head without disrupting this structure.
>
> ---
>
> **W2: MoE framework lacks conceptual depth; PCA overlaps with prior work; no causal evidence linking QK restriction to degradation.**
>
> We respectfully argue that the MoE analogy provides genuine conceptual value, and that our analysis contributes a unified comparative perspective.
>
> * **On conceptual depth.** By paralleling expert collapse, we identify head collapse as an analogous training pathology and motivate load balancing as a remedy. This lens also reinterprets head sparsity for inference acceleration and suggests head-based parameter scaling analogous to MoE, which we plan to explore in future work.
>
> * **On the PCA analysis.** We agree that attention sinks reducing effective rank has been noted before, and we cite it. Our contribution is a **unified comparative analysis across all three mechanisms** to explain their expressive gap.
>
> * **On causality.** We propose the chain: **attention sink → restricted QK space → attention noise → performance degradation.** The first link is shown by our experiments; the second is supported by prior work on attention noise [1][2][3], where restricted QK space causes misallocation to irrelevant tokens; the third is evidenced in Table 1 and Figure 7 of the original paper. Ranking mechanisms is not our goal; prior work [1] has addressed that.
>
> ---
>
> **Q1: Does the auxiliary loss interfere with beneficial attention properties, e.g., retrieval head identification?**
>
> In short, retrieval heads remain identifiable, and the loss encourages richer specialization.
>
> * **Retrieval heads are preserved.** Following DuoAttention [4], we apply the same head classification to our fine-tuned models. **As shown in Figure 5 (attached pdf), retrieval and streaming heads remain clearly identifiable**, confirming that structural attention properties are preserved.
>
> * **Finer-grained specialization emerges.** Beyond the binary taxonomy, **Figure 6 (attached pdf) shows that different heads develop clearer task-level preferences**: some are more active for knowledge (MMLU), math (GSM8K), commonsense (PIQA), or code (HumanEval). This mirrors expert specialization in MoE.
>
> * **Sequence-level patterns are more structured.** Figure 7 (attached pdf) shows that with the auxiliary loss, head activations are distributed more evenly across tokens while individual heads retain distinct profiles, consistent with the MoE analogy.
>
> ---
>
> **References:**
>
> [1] Gated Attention for Large Language Models
>
> [2] Differential Transformer
>
> [3] gpt-oss-120b & gpt-oss-20b Model Card
>
> [4] DuoAttention: Efficient Long-Context LLM Inference with Retrieval and Streaming Heads
>
> ---
>
> **We thank the reviewer for the valuable suggestions and will incorporate these discussions into the revised version.**

---

> > ### Author Rebuttal · Reviewer_QbW5 · 2026-04-06
> >
> > Thank you to the authors for the additional experiments and detailed responses. My concerns have been addressed, and I now believe the paper is solid both theoretically and empirically. Therefore, I have increased my score.

---

> > > ### Author Response · Authors · 2026-04-06
> > >
> > > We are genuinely pleased that our response has addressed your concerns, and we sincerely appreciate your recognition of this work.
> > >
> > > Your feedback on the connection between *QK restriction* and *performance degradation* led us to a clearer and more rigorous narrative.
> > > Furthermore, your question regarding *beneficial attention properties* prompted us to conduct the retrieval head analysis, which revealed the *richer task-level specialization* that emerges with our auxiliary loss. This is a meaningful finding that strengthens the paper.
> > >
> > > Should the paper be accepted, **we will include these additional discussions in the final version**.
> > >
> > > **Thank you again for your effort and the thoughtful engagement throughout this review process!**

---

### Official Review · Reviewer_uG1u · 2026-03-12

**Soundness:** 3
**Presentation:** 3
**Significance:** 3
**Originality:** 3
**Overall Recommendation:** 4
**Confidence:** 4

**Summary:**

This paper presents a novel theoretical insight connecting the "attention sink" phenomenon in LLMs to a native MoE structure within attention layers. The authors demonstrate that the attention sink functions as an implicit gating mechanism, which is mathematically equivalent to the explicit gating in recently proposed Gated Attention. Building on this perspective, the paper identifies a critical issue: head collapse, where only a small subset of attention heads are actively utilized, leading to imbalanced head utilization and suboptimal model capacity. To address this, the authors propose a sink-aware auxiliary load balancing loss that encourages uniform head utilization during training. Extensive experiments on models ranging from 0.6B to 2B parameters trained from scratch, as well as fine-tuning on Qwen3 and LLaMA3.1 models (up to 8B), demonstrate that the proposed loss consistently improves head balance and boosts performance across a wide range of benchmarks.

**Compliance With Llm Reviewing Policy:**

Affirmed.

**Final Justification:**

This paper investigates the common phenomenon of attention sink and proposes a new method based on its findings. The contribution is significant, and the rebuttal has addressed the concerns. Therefore, I recommend accepting this paper.

**Key Questions For Authors:**

See weakness above.

**Limitations:**

Yes.

**Strengths And Weaknesses:**

Strength：
(1) This paper reframes attention sink as a fundamental feature that forges an implicit MoE structure, providing a fresh and unifying perspective on attention mechanisms. This conceptual contribution is significant and thought-provoking.
(2) The paper provides clear mathematical derivations showing the equivalence between the implicit gating from attention sink and the explicit gating in Gated Attention.
(3) The paper presents extensive results across multiple model scales (0.6B, 1B, 2B), three attention mechanisms, and a wide variety of downstream tasks. The fine-tuning experiments on Qwen3 and LLaMA3.1 further validate the method's applicability to existing models.

Weakness:
(1) The insight behind the design of the auxiliary loss could be presented more clearly. For example, what is the benefit of this particular design? Could it be designed to maximize the importance of each head individually, rather than making the importance of different heads balanced?

---

> ### Author Rebuttal · Authors · 2026-03-30
>
> We sincerely thank Reviewer uG1u for the careful reading and constructive feedback. We are glad the theoretical contributions and experimental breadth were found compelling. Below we address the raised concern in detail.
>
> **Please see the newly added experimental results at [https://anonymous.4open.science/api/repo/ICML-2026-395/file/ICML_2026_response.pdf](https://anonymous.4open.science/api/repo/ICML-2026-395/file/ICML_2026_response.pdf).**
>
> ---
>
> **W1: Insight behind the design of auxiliary loss**
>
> **Core insight.** Our paper establishes that the attention sink functions as an implicit gating mechanism across three attention variants, forging a native MoE structure within attention layers. This naturally raises a question: if attention heads behave like MoE experts, do they suffer from load imbalance? We show they do, a phenomenon we term *head collapse*, where only a small subset of heads are consistently activated while others remain dormant. **The auxiliary load balancing loss is designed to address this structural inefficiency by promoting balanced head utilization, directly leveraging the attention sink as the gating signal.**
>
> * **Design choices.** Two families of auxiliary balancing losses exist in the MoE literature. The frequency-based approach [2] penalizes $f_i \cdot p_i$, where $f_i$ is the token-level activation frequency and $p_i$ is the routing probability. This requires explicit top-$k$ routing and discrete frequency computation, which are unavailable in standard attention. More critically, since head gating factors are unnormalized across heads, minimizing this objective would drive all gating factors toward zero, worsening head collapse rather than alleviating it.
>
> * We instead adopt a CV-based formulation [1]:
> Since CV normalizes by the mean: $\mathrm{CV}(\cdot) = \frac{\mathrm{std}(\cdot)}{\mathrm{mean}(\cdot)}$, **only relative differences across heads matter; the absolute magnitude of gating factors is irrelevant.** The loss therefore neither amplifies nor suppresses the attention sink phenomenon — it uses the sink purely as a gating signal to encourage balanced head utilization.
>
> * **Sensitivity to $\lambda$.** Figure 11 (attached pdf) shows that our loss consistently improves validation BPB and downstream accuracy across a wide range of $\lambda$. An excessively large $\lambda$ can interfere with the primary language modeling objective [2], so we adopt $\lambda = 10^{-4}$ throughout all main experiments.
>
> * **Adaptation for fine-tuning.** Applying the loss directly to pretrained models triggers a *head pinning effect*: dominant collapsed heads resist rebalancing, causing other heads to inflate their gating factors rather than achieving genuine balance. Inspired by DeepSeek-MoE, we designate the top-$m$ heads per layer as *shared heads* and apply balancing only to the remaining *routed heads*, enabling stable fine-tuning without disrupting well-learned behaviors. A detailed analysis is provided in Appendix D.
>
> ---
>
> **W1 (cont.): Why balance rather than maximize each head individually?**
>
> This is a valid alternative; **however, we argue that balancing head importance across heads is the more principled choice.** We provide the following reasons:
>
> * **It is well-established in the MoE literature [1, 2] that beneficial sparsity should be preserved, not eliminated.** Sparse, input-dependent expert activation reduces redundancy and encourages functional specialization. The goal of load balancing is not to make every expert equally active on every token, but to ensure no expert is permanently ignored.
>
> * **Our experiments support this design.** Figure 6 (attached pdf) shows that after applying our auxiliary loss, heads develop clear task preferences: Head 12 is more active on academic knowledge (MMLU), Head 6 on mathematical reasoning (GSM8K), Head 10 on physical commonsense (PIQA), and Head 0 on code generation (HumanEval), closely resembling expert specialization in MoE models. Without our loss, certain heads (e.g., Heads 0, 5, 12) remain permanently inactive regardless of input, wasting model capacity. Figure 7 (attached pdf) further confirms that, at the sequence level, heads exhibit more varied and input-dependent activation patterns with our loss, with overall utilization more evenly distributed across heads.
>
> **In summary, our auxiliary loss prevents pathological head collapse while preserving the beneficial input-dependent sparsity that makes the implicit MoE structure of attention layers effective.**
>
> ---
>
> **References**
>
> [1] Outrageously Large Neural Networks: The Sparsely-Gated Mixture-of-Experts Layer
>
> [2] Switch Transformers: Scaling to Trillion Parameter Models with Simple and Efficient Sparsity
>
> ---
>
> **We thank the reviewer again and will incorporate these clarifications into the revised paper.**

---

> > ### Author Rebuttal · Reviewer_uG1u · 2026-04-04
> >
> > Thanks for your response. My concerns have been addressed.

---

> > > ### Author Response · Authors · 2026-04-04
> > >
> > > We are glad to hear that our response has addressed your concerns! If you find the contributions of this work valuable, we would be grateful for your consideration in updating your score accordingly.
> > >
> > > The clarification on the *insight behind the design of the auxiliary loss* was particularly valuable in making the paper's motivation and technical narrative clearer, and **the relevant discussion will be incorporated into the final version** upon acceptance.
> > >
> > > **Thank you again for your time and the insightful exchange throughout this review process!**

---

### Decision · Program_Chairs · 2026-04-30

**Decision:**

Accept (regular)

**Comment:**

This paper provides a novel perspective by demonstrating that "attention sinks" in LLMs function as an implicit gating mechanism, effectively forming a native MoE structure within standard attention layers. Building on this theoretical framework, the authors identify the "head collapse" phenomenon and propose a simple yet highly effective sink-aware load-balancing loss to optimize head utilization. All reviewers unanimously praised the paper's conceptual insights, clear mathematical grounding, and extensive empirical validation across multiple model scales. My recommendation is acceptance.